# SARS-CoV-2 Evolutionary Adaptation toward Host Entry and Recognition of Receptor O-Acetyl Sialylation in Virus–Host Interaction

**DOI:** 10.3390/ijms21124549

**Published:** 2020-06-26

**Authors:** Cheorl-Ho Kim

**Affiliations:** 1Molecular and Cellular Glycobiology Unit, Department of Biological Sciences, Sungkyunkwan University, Suwon 16419, Korea; chkimbio@skku.edu; 2Samsung Advanced Institute for Health Sciences & Technology (SAIHST), Sungkyunkwan University, Seoul 06351, Korea

**Keywords:** SARS-CoV-2, evolutionary adaptation, entry receptor, sialic acid O-acetylation, hemagglutinin-esterase, glycosylation, virus–host interaction

## Abstract

The recently emerged SARS-CoV-2 is the cause of the global health crisis of the coronavirus disease 2019 (COVID-19) pandemic. No evidence is yet available for CoV infection into hosts upon zoonotic disease outbreak, although the CoV epidemy resembles influenza viruses, which use sialic acid (SA). Currently, information on SARS-CoV-2 and its receptors is limited. O-acetylated SAs interact with the lectin-like spike glycoprotein of SARS CoV-2 for the initial attachment of viruses to enter into the host cells. SARS-CoV-2 hemagglutinin-esterase (HE) acts as the classical glycan-binding lectin and receptor-degrading enzyme. Most β-CoVs recognize 9-*O*-acetyl-SAs but switched to recognizing the 4-*O*-acetyl-SA form during evolution of CoVs. Type I HE is specific for the 9-*O*-Ac-SAs and type II HE is specific for 4-*O*-Ac-SAs. The SA-binding shift proceeds through quasi-synchronous adaptations of the SA-recognition sites of the lectin and esterase domains. The molecular switching of HE acquisition of 4-*O*-acetyl binding from 9-*O*-acetyl SA binding is caused by protein–carbohydrate interaction (PCI) or lectin–carbohydrate interaction (LCI). The HE gene was transmitted to a β-CoV lineage A progenitor by horizontal gene transfer from a 9-*O*-Ac-SA–specific HEF, as in influenza virus C/D. HE acquisition, and expansion takes place by cross-species transmission over HE evolution. This reflects viral evolutionary adaptation to host SA-containing glycans. Therefore, CoV HE receptor switching precedes virus evolution driven by the SA-glycan diversity of the hosts. The PCI or LCI stereochemistry potentiates the SA–ligand switch by a simple conformational shift of the lectin and esterase domains. Therefore, examination of new emerging viruses can lead to better understanding of virus evolution toward transitional host tropism. A clear example of HE gene transfer is found in the BCoV HE, which prefers 7,9-di-*O*-Ac-SAs, which is also known to be a target of the bovine torovirus HE. A more exciting case of such a switching event occurs in the murine CoVs, with the example of the β-CoV lineage A type binding with two different subtypes of the typical 9-*O*-Ac-SA (type I) and the exclusive 4-*O*-Ac-SA (type II) attachment factors. The protein structure data for type II HE also imply the virus switching to binding 4-O acetyl SA from 9-O acetyl SA. Principles of the protein–glycan interaction and PCI stereochemistry potentiate the SA–ligand switch via simple conformational shifts of the lectin and esterase domains. Thus, our understanding of natural adaptation can be specified to how carbohydrate/glycan-recognizing proteins/molecules contribute to virus evolution toward host tropism. Under the current circumstances where reliable antiviral therapeutics or vaccination tools are lacking, several trials are underway to examine viral agents. As expected, structural and non-structural proteins of SARS-CoV-2 are currently being targeted for viral therapeutic designation and development. However, the modern global society needs SARS-CoV-2 preventive and therapeutic drugs for infected patients. In this review, the structure and sialobiology of SARS-CoV-2 are discussed in order to encourage and activate public research on glycan-specific interaction-based drug creation in the near future.

## 1. Introduction

The recent coronavirus pandemic crisis is due to viral infection of severe acute respiratory syndrome-related coronavirus-2 (SARS-CoV-2), causing uncontrolled inflammatory conditions in the human lung. Soon after the first transmission emergence of SARS-CoV from animals to humans in China in 2003 [1], a genetically evolved beta-coronavirus genus similar to human viruses was discovered in Chinese horseshoe bats (*Rhinolophus sinicus*) [2]. To date, pneumonia is epidemiologically caused by diverse viruses. For example, adenovirus, influenza virus, Middle East respiratory syndrome virus (MERS-V), parainfluenza virus, respiratory syncytial virus (RSV), SARS-CoV and enteric enveloped CoV can cause pneumonia in human hosts.

The World Health Organization (WHO) reported the official terminology of the 2019-Novel Coronavirus (2019-nCoV) on 13 January 2020, and on February 11th, WHO edited the name of the disease caused by 2019-nCoV to Coronavirus Disease-2019 (COVID-19). In academia, the International Committee on Taxonomy of Viruses (ICTV) provided official nomenclature to the virus as SARS-CoV-2 due to the similarity between the novel coronavirus and SARS-CoV [3]. SARS-CoV-2 is spreading and causing a global health-threatening emergency [4]. Researchers have been in a race to develop anti-viral drugs against SARS-CoV-2 even before the WHO declared a worldwide pandemic threatening human lives. Preventive and therapeutic drugs for patients infected with SARS-CoV-2 are yet to be discovered.

The CoVs as enveloped forms can also infect the gastrointestinal track (GIT), although most other enteric viruses are naked in morphology [5,6]. CoVs can also rarely infect neural cells [7]. There is, unfortunately, no solid information on how the coronaviruses infect humans and animals with reciprocal infectivity and cause a zoonotic viral outbreak. This is in contrast to influenza viruses, which are known to selectively utilize sialic acid (SA) linkages [8]. Currently, only limited information is available on β-CoVs, such as SARS-CoV and its receptor usage and infectible cell types from different species. Host cell surface O-acetylated SAs are recognized by the lectin-like spike proteins of SARS CoV-2 for the first step of attachment to host cells. Infectious virus interaction with the host cell surface is mediated by sialoglycans as the most important phenomenon in eukaryote-parasite co-evolution. O-GlcNAc, a minor glycan source, is mainly found in the nucleus and cytosol (Figure 1). Apart from the general roles of glycans, CoVs recognize host cells and attach to host cell surface molecules to enter the host cells. For example, activity of the hemagglutinin-esterase (HE) enzyme relies on the typical carbohydrate-binding lectin and receptor-destroying enzyme (RDE) domains. Most β-CoVs target 9-*O*-acetylated SAs, but certain species have switched to recognizing 4-*O*-acetyl SA instead [8,9]. Crystallographic data for the molecular structure of type II HE provides an explanation for the switching mechanism to acquire 4-O acetyl SA binding. This event follows the orthodox ligand–receptor interaction (LRI), lectin–carbohydrate interaction (LCI), lectin–glycan interaction (LGI), lectin–sphingolipid interaction (LSI), protein–glycan interaction (PGI), protein–carbohydrate interaction (PCI), and also protein–protein interaction (PPI). Recently, 332 protein candidates were suggested to be SARS-CoV-2-human protein interacting proteins through PPIs. Among these, 66 human proteins, as druggable host factors, were further characterized as possible FDA-approvable drugs [10]. If PPI is involved, however, carbohydrates or glycans may serve as co-receptors or co-determinants. Previous reports suggest that carbohydrates act as receptor determinants in most cases. The general principles of PCI stereochemistry potentiate the SA–ligand switch by way of simple conformational shifts for the lectin and esterase domain. This indicates that our examination of natural adaptation should be directed to how carbohydrate-binding proteins measure and observe carbohydrates, leading to virus evolution toward transitional host tropism.

The 9-carbon SAs are mainly animal-specific with anionic sugars attached to terminal sugars. SAs exist in two forms, NeuGc and NeuAc. NeuGc is a differentially modified form of the parental SA form, Neu5Ac. SAs are structurally diverse. For example, several modified SA forms are known for their structures including neuraminic acid (NeuC), N-acetyl neuraminic acid (NeuAc), N-glycolyl neuraminic acid (NeuGc), N,O-diacetyl neuraminic acid (occurs in horses), N,O-diacetyl neuraminic acid (occurs in bovines) and N-acetyl O-diacetyl neuraminic acid (occurs in bovines) (Figure 2). Sialyltransferases (STs) biosynthesize different SA linkages. SA linkage diversity occurs at the α2-3, α2-6, α2-8 or α2-9 to the SA or Gal residues (Figure 3). For example, in formation of α2,3 SA or α2,6 SA structures, α2,3-ST and α2,6-ST utilize substrates such as Galβ-1,4-GlcNAc (Figure 4). The most frequent modification of SAs is O-acetylation at positions of C4, C7, C8 and C9 of SA (Figure 5).

## 2. Classification of CoVs

Mammal- and avian-infectible CoVs consist of the broad-ranged subfamily of Coronavirinae. The ICTV recommended classification is as follows: Riboviria (Realm)—Nidovirales (Order)— Cornidovirineae (Suborder)—coronavirida (Family)—orthocoronavirinae—*Coronavirus* genus. The four CoV genera [11] are the α-CoV, β-CoV, γ-CoV and δ-CoV. The 2019-nCoV or SARS-CoV-2 belong to the β-CoV genus and are zoonotic and cause mammalian infection, causing respiratory disease in the human lung. The α-CoV and β-CoV genus target mammal hosts while the δ-CoV and γ-CoV genus target avians and certain mammals. The β-CoV genus has A, B, C and D lineages. Among these, lineage B includes SARS-CoV and SARS-CoV-2. Lineage C includes MERS-CoV. The B lineage SARS-CoV and C lineage MERS-CoV, which are classified as β-CoVs, exhibit lethal rates of 10% and 35% in humans, respectively.

CoVs in humans are associated with respiratory infections such as colds with clinical importance, as experienced for the previous outbreak in 2003 of SARS-human CoV (HCoV), HCoV-HKU1 and HCoV-NL63 [12,13]. Human infectious HCoV includes seven species, including α-CoV (HCoV-NL63 and HCoV-229E) and β-CoV (SARS-CoV, SARS-CoV-2, HCoV-OC43, HCoV-HKU1 and MERS-CoV). CoV RNA sequences mutate at a high frequency. Among the known RNA viruses, CoVs bear the longest genome sizes of 26 to 32 kb length RNA. Nucleotide sequences of CoV ssRNA genomes isolated from COVID-19 patients in Wuhan show a high homology of 89% with the nucleotide sequence of the previously known bat SARS-like CoV-ZXC-21 strain and 89% with the previous SARS-CoV. The initial Wuhan CoV isolates belong to the β-CoV genus and were therefore termed SARS-CoV-2 or 2019-nCoV [14]. SARS-CoV-2 infects human respiratory tracts and causes outbreaks of pneumonia. SARS-CoV-2 is a novel CoV and originates from the Wuhan district in China. The genome sequence of SARS-CoV-2 exhibits 79% sequence homology with the SARS-CoV RNA sequence and 50% with the MERS-CoV sequence [15].

## 3. Structure, Components and Life Cycle of CoVs

CoVs are 60–140 nm in size and are enveloped (+) ssRNA viruses, which feature an RNA genome, directly available to function as mRNA and thus result in rapid infection. CoVs exhibit RNA genomes of 28–32 kb, comprised of two large overlapping open reading frames (ORFs), which encode the virus replicase (transcriptase) and structural proteins. The SARS-CoV-2 genome is 29,891 bp in size, which encodes 9860 amino acids. The ssRNA are capped and tailed with a 5′-capping structure and 3′-poly A tail at the termini. The genome is the same sense as virus mRNA indicating that the viral RNA is translated through its own (+) RNA to synthesize RNA dependent RNA polymerase (RdRp; PDB: 6M71).

Generally, viral families are determined by the genome structure and virion morphologies of an envelope or naked capsid. A virus with a naked capsid has a coat of nucleocapsid protein (N) coating the viral genome. Viruses with an envelope have lipid envelopes further surrounding the outmost protein layer. The 2019-nCoV (SARS-CoV-2) contains a spike (S) glycoprotein, E, dimeric HE enzyme, a membrane matrix glycoprotein (M), N and RNA [16]. The structural proteins are the S, N, M and E proteins, while the non-structural proteins are proteases such as Nsp3 and Nsp5 and RdRp such as Nsp12. Among the N, M and S glycoproteins, the S glycoprotein is a fusion protein that recognizes the host receptor and enters the host cells [17]. The S, M and E proteins anchored into the endoplasmic reticulum (ER) membrane are trafficked to the endoplasmic reticulum–Golgi intermediate compartment (ERGIC). The RNA genome linked with nucleoprotein buds into the ERGIC to form virus particles. Assembled virions transported to the vesicular surface are released to the extracellular milieu via exocytosis.

The RNA generates the replicase as two polyproteins, pp1a and pp1ab. The replicase-encoded viral proteases generate up to 16 nonstructural proteins (Nsps) in the cytosol to produce replicase enzyme and the replicase–transcriptase complex (RTC). These enzymes including RTC synthesize RNAs for replication and transcription to generate viral RNA genome. CoV genomes bear two or three protease genes and the coding enzymes cleave the replicases. Together with the replicases, nonstructural proteins, termed Nsps, assemble into the RTC complex. Nsp1 to Nsp16 are known to have multiple enzyme regions. For example, Nsp1 degrades cellular mRNAs and, consequently blocks protein translation in host cells and innate immune responses. Nsp2 recognizes the specific protein called prohibitin. Nsp3 is a multi-domain transmembrane (TM) protein with diverse activities. Ubiquitin-like 1 and acidic domains bind to N protein and ADP-ribose-1′-phosphatase (ADRP) activity induces cytokine expression. The papain-like protease (PLpro)(PDB:6WX4)/ deubiquitinase domain cleaves virus-produced polyprotein. Nsp4 is a TM scaffold protein for double-membrane vesicle structure. Nsp5 has a main protease domain which also cleaves virus-produced polyprotein and Nsp6 acts as a TM scaffold protein. The Nsp7 and Nsp8 proteins form the Nsp7-Nsp8 hexadecameric complex, which functions as an RNA polymerase-specific clamp and a primase enzyme. Nsp9 is an RNA-binding protein that activates ExoN and 2-*O*-methyltrnasferase (MTase) enzyme activity. Nsp10 binds to Nsp16 and Nsp14 to form a heterodimeric complex. Nsp12 is the RdRp and Nsps13 is the RNA helicase and 5′ triphosphatase. Nsps14 is a N7 MTase and 3′-5′ exoribonuclease. ExoN of Nsap14 acts as an N7 MTase and attaches the 5′ cap to viral RNAs. Viral exoribonuclease enzyme proofreads the viral RNA genome, where Nsp15 is a viral endoribonuclease (PDB: 6VWW). Nsp16 has 2-*O*-MTase enzyme activity, which shields viral RNA from melanoma differentiation associated protein-5 recognition.

### 3.1. Spike (S) Transmembrane Glycoprotein

In RNA viruses, the S glycoprotein (PDB: 6VSB) is the biggest protein, heavily glycosylated and its N-terminal domain (NTD) sequence binds to the host receptor to enter the ER of host cells. SARS-CoV-2 S-glycoprotein bears 22 N-glycan sequons in each protomer. Therefore, the trimeric S glycoprotein surface is dominated by 66 N-glycans. The S glycoprotein mediates direct and indirect interaction of virus with host cells in the infection cycle. All CoVs exhibit a surface S glycoprotein, which bears the receptor-binding domain (RBD). The S glycoprotein has a distinct spike structure. When S glycoprotein binds to its host receptor, a host furin-like protease cleaves the S glycoprotein, which liberates the spike fusion peptides, allowing entry of the virus into the host cell [18]. The furin-like protease-generated S1 and S2 exist as a S1/S2 complex, where S1 in a homotrimeric form interacts with the host cell membrane and S2 penetrates the cytosolic area. For SARS-CoV and MERS-CoV, the S1 C-terminal domains (CTDs) have a dual role in virus entry via attachment and fusion. The S1 NTD binds to carbohydrate receptors because the S1 domains act as the RBD. The CTD of S1 recognizes protein receptors via RBDs.

### 3.2. Nucleocapsid (N) Protein

In RNA viruses, the N protein recognizes the viral RNA genome. The N protein (PDB: 6M3M) binds to the RNA genome via the NTD and CTD. The N protein tethers to the viral RNA and replicase–transcriptase complex (RTC). NSP3 (PDB: 6VXS) of CoV blocks the innate immune responses of hosts. After entrance into the host cells, for CoV transcription and particle release, RNA chaperones such as nonspecific nucleic acid binding proteins potentiate ssRNA conformation shifts. Representatively, the N protein is known as the RNA chaperone protein. For example, glycogen synthase kinase 3 (GSK3) phosphorylates the SARS-CoV N-protein and thus, GSK3 inhibition contributes to reduced replication activity of SARS-CoV [19]. In addition, heterogeneous nuclear ribonucleoprotein A1 (hnRNP A1) regulates the preformed mRNA splicing in the nucleus and continuous translation. hnRNPA1 interacts with SARS-CoV N protein to form a replication and transcription complex during ssRNA genome biosynthesis [20].

### 3.3. Envelope (E) Protein

E protein is the most abundant structural protein needed to assemble virus particles in the cytosol. As a TM protein, E protein is the smallest structural protein with a MW of 12 kDa. The E protein has an NTD in the extracellular region and a CTD in the cytosolic region. The E protein bears an ectodomain in the NTD and an endodomain in the CTD. It has also an ion channel domain. E protein is present in the cytosolic region of infected cells and only a limited amount is incorporated into the envelope of virions [21]. Most E proteins assemble and bud in new virus particles.

### 3.4. Membrane Glycoprotein (M)

The M protein contains three TM domains and is an abundant structural protein with a MW of 30 kDa. It consists of a small glycosylated NTD in the extracellular region and CTD in the cytoplasmic region. The M protein forms a scaffold for virus assembly in the cytosol via binding to S glycoprotein and N protein [22]. For example, E protein and N protein are co-expressed with M protein to form virus-like particles (VLPs) that are released from the cells, as the M and E protein are involved in CoV assembly. Then, CoVs bud into the ERGIC, trafficking by membrane vesicles and transported via the exocytosis-secretory pathway [23,24]. The dimeric M protein binds to the nucleocapsid.-M protein binds to S glycoprotein for S glycoprotein retention in the ERGIC/Golgi complex. The M-N protein complex keeps the N protein–RNA complex stable, for nucleocapsid and the viral assembly.-M and E proteins constitute the virus envelope for successful release of virus-like particles.

### 3.5. Hemagglutinin-Esterase (HE) Dimeric Protein

HE hemagglutinates and destroys receptors. As RNA viruses, CoVs bear RDEs, which are used in effective attachment to hosts and also reversely in detachment from the hosts. For example, enveloped RNA viruses evade the hosts via their RDEs. Currently, RDE-related functional enzymes such as neuraminidase (NA) and SA-*O*-acetyl-esterase are known. SA-*O*-acetyl-esterase was originally identified in influenza C virus and in nidoviruses (CoV and torovirus) as well as in salmon anemia virus (teleost orthomyxovirus). The origin and evolution of CoV SA-*O*-acetylesterases are correlated to other viruses. The fusion event of S glycoprotein and HE is specific for HCoV attachment to SA-associated receptors in the host [25]. The HE has acetylesterase activity [26]. In early SA-related biology, influenza A/B viruses were found to recognize chicken erythrocytes in 1942 [27]. They caused hemagglutination through clumping by virus-borne hemagglutinin. These phenomena were widely found in influenza viruses, paramyxoviruses, Newcastle disease (NDV) and mumps virus.

## 4. Relationship between C4-*O*- and C9-*O*-Acetyl SA Preferences of CoVs in Host Cell Recognition

### 4.1. General O-Acetylation of SA

SAs have various derivatives of more than 50 chemically different structures formed from the basic N-acetylneuraminic acid (Neu5Ac) on the main ring of pyranose and the glycerol side chain. SA are modified by acetyl-, lactyl-, methyl- and sulfo-groups individually or in multiple combinations [28]. Multiple enzymes are involved in the modifications [29]. Historically, the first discovered SA was crystallized by Gunner Blix via a hot mild acid extraction of bovine submaxillary mucin in 1936. It consisted of two acetyl groups. Among these, only one acetyl group was attached to nitrogen [30]. Blix isolated a 9-*O*-acetyl SA of the common SA N-acetylneuraminic acid (Neu5Ac), chemically described as 9-*O*-acetyl-N-acetylneuraminic acid (Neu5,9Ac2). Neu5,9Ac2, Neu5Ac and Neu5Gc are naturally occurring SA species in mammals. A common modification is O-acetylation. In fact, O-acetylation of SAs is common in organisms.

The O-acetyl modification occurs in single positions of C-4, C-7, C-8 and C-9 of SA as well as in combined C-positions to yield Neu4,5Ac2, Neu5,9Ac2 and Neu5,7,9Ac3 SAs. Neu5Ac9NAc is a chemical and biologic mimic of Neu5,9Ac2 in the SA-glycans. The C-7 and/or C-9 O-acetylations are catalyzed by the SA O-acetyltransferase enzyme, Cas1 domain containing 1 (CasD1) (Figure 6). CasD1 catalyzes the addition of acetyl groups to the SA C-7 at the late Golgi apparatus compartment [31]. Thereafter, an enzyme termed “migrase” transfers the additional acetyl-group from C-7 to C-9, although this enzyme has not been identified [29]. CasD1 uses acetyl-coenzyme A as a donor substrate and CMP-Neu5Ac as an acceptor substrate, but with weak activity on CMP-Neu5Gc.

Biologically, the SA O-acetylation event confers merits to hosts such as protection from pathogenic invasion and maintenance of systemic self-homeostasis. The O-acetylation event of SAs protects the SA-containing glycans from neuraminidase (NA)/sialidase action, because O-acetyl-groups inhibit microbial NA activity. The chemical structure of the O-acetyl group is quite unstable and susceptible to esterase enzymes. Sialic acid cleavage of the di-acetylated Neu5,7,9Ac3 by bacterial NAs decreases two-fold, when compared to mono-*O*-acetylated Neu5Ac. The O-acetylated glycan modification invites interaction with viruses, antibodies and mammalian lectins [32]. Therefore, the SA O-acetylation modification confers specific functions to organisms.

### 4.2. Evolutionary Acquisition of C4-*O*-Acetyl and C9-*O*-Acetyl SA Recognition by HE Enzymes

#### 4.2.1. C4-*O*-Acetyl Modification

For example, in the horse, C4-*O*-acetyl modification of Neu5Ac (SA) occupies more than 50% of the total SA content. The C4-*O*-acetylated Neu5Ac, Neu4,5Ac2, inhibits the influenza A2 virus HA. De-acetylation reagents such as NaOH or NaIO_4_ treatment completely hemagglutinate Neu4,5Ac2 by elimination of the C4-*O*-acetyl group [33]. The C4-*O*-acetyl Neu5Ac species are found in various sources such as equine erythrocyte GM3, starfish *A. rubens* and fish [34,35,36,37,38]. C4-*O*-acetylated Neu5Ac facilitates the initial attachment of viruses to target cells. Like the influenza C virus, infectious salmon anemia virus (ISAV), a member of the Orthomyxoviridae family, contains HE and HEF proteins to mediate virus entry and exit. C4-*O*-Ac Neu5Ac is the major receptor determinant of ISAV in receptor binding and destruction [38], while the influenza C virus recognizes C9-*O*-Ac Neu5Ac. The acetylesterase RDE of ISAV cleaves C4-*O*-Ac via 4-SA-*O*-acetylesterase with a short turnover time, whereas C9-*O*-Ac Neu5Ac is cleaved by 9-SA-*O*-acetylesterase with a long turnover time [34].

The position of SA O-acetylation is linked to functions including substrate differentiation of enzymes such as NAs and esterase by C4 O-Ac. Previous development of O-Ac site-selective NA inhibitors were based on the conceptual consideration of different O-Ac positions. The O-Ac of SAs is site-specific, as C4 of Neu5Ac is considered to be a potential position for modification. Historically, inhibitors of influenza A and B viruses-sialidases were designed by Von-Itzstein in 1993 [39]. The Ac group-based C4 substitution interacts with amino acid Glu-119 present in the active site of sialidase. Guanidine-attached C4 of C2–C3 unsaturated SA (Neu5Ac2en) inhibits activity of sialidases isolated from influenza A virus (Singapore/1/57) and B virus (Victoria/102/85). The same scenario was applied for sialidase inhibition of the human parainfluenza virus type 3, which has HN and fusion proteins [40]. The C4 of Neu5Ac2en was substituted by alkyl groups such as the O-ethyl group. For example, Zanamivir has a substitution with a 4-guanidino group with an IC50 of 25 μM. Thus, sialidase inhibition is important for C4 modification of Neu5Ac2en. Later, oseltamivir with the tradename Tamiflu (Basel, Switzerland) and zanamivir with the tradename Relenza (London, UK) were established [41]. These drugs exhibit some adverse side effects that restrict clinical use.

#### 4.2.2. SA C9-*O*-Acetyl Modification

The SA 9-*O*-acetylation in hosts allows hosts to evade influenza A virus hemagglutinin (HA) recognition and some lectins of factor H (FH), CD22/Siglec-2 and sialoadhesin/Siglec-1. Instead, the influenza C virus HA recognizes the hosts. β-elimination and permethylation eliminate the 9-*O*-acetyl group from SAs. Chemical modification of the C-9 position of Neu5,9Ac2 generates a 9-N-acetyl analog, 9-acetamido-9-deoxy-N-acetylneuraminic acid (Neu5Ac9NAc), a mimic of Neu5,9Ac2 with influenza C virus-binding capacity, which is not cleaved by the HE [42]. SA O-acetylesterase regulates the presence of 7,9-*O*-Ac and 9-*O*-Ac. SA O-acetylation and deacetylation are involved in development, cancer and immunology. SA O-acetylation alters host lectin bindings such as siglecs [29]. The presence of 9-*O*-Ac can also reduce the activity of NAs [43]. SA modifications regulate pathogen binding or pathogen NAs. Influenza A/B/C/D viruses use SA as their entry receptors. Influenza A and B subtypes bind to SAs via HA and NA to allow endocytosis of the virus and fusion of the viral envelope with endosomes. In contrast, influenza C and D subtypes bear only one coated glycoprotein, termed the HE fusion protein (HEF). The HEF acts as the HA and NA. HEF recognizes 9-*O*-acetyl SA for entry into cells, while the esterase domain removes 9-*O*-acetyl-groups and liberates the virus from mucus and mis-assembled virus aggregates after budding. The 9-*O*-Ac on cells prevents the NA activity and HA binding of the influenza A type virus [44].

## 5. HE of CoVs

### 5.1. Evolutionary Origin and Classification of the CoV HE

Certain viruses use glycoproteins such as HA, HE, S and HEF for host receptor binding or destruction. Coronaviridae, Orthomyxoviridae, Paramyxoviridae and Adenoviridae utilize SAs as binding molecules for attachment and entry. However, only limited human pathogens recognize O-Ac SA.

#### 5.1.1. Influenza Virus A and B Spike Proteins of HA and NA

Influenza A and B viruses bear two spikes of receptor-binding HA and NA [45].

#### 5.1.2. Influenza C virus HA-HEF

HEF is indeed an ancient type of SA-*O*-acetylesterase. In contrast to A/B, the influenza C virus bears one spike with triple functions of HEF as a homotrimer [46]. Each HEF subunit bears two Neu5,9Ac2-binding sites and binds to the 9-*O*-acetyl group. In parallel, another modification of O-acetylation is found. Indeed, influenza C virus bears SA-*O*-acetylesterase [47], which converts 5-N-acetyl-9-*O*-NeuAc (Neu5,9Ac2) to 5-Neu5Ac. The 9-*O*-acetyl SA is a unique determinant for the influenza C virus receptor and Neu5,9Ac2 is crucial for receptor activity, but not Neu5Gc or Neu5Ac [48]. Neu5,9Ac2 is an essential determinant for influenza virus C type-specific host cell tropism. NAs cleave the α-ketosidic linkages to the D-Gal or GalNAc. SA-*O*-acetylesterases cleave different O-acetyl linkages (Figure 5). The OH-group of Tyr224 and the guanidino group of Arg236 interact with the CH_3_CO-carbonyl oxygen [49]. HEF SA-*O*-acetylesterase is found in several enveloped (+) ssRNA viruses of influenza C virus and also in certain CoVs and toroviruses [47]. The CoVs are different from the orthomyxoviruses, which hold a segmented (−) ssRNA genome and are instead evolutionary linked to the family Coronaviridae, order Nidovirales [45].

#### 5.1.3. CoV SA-*O*-Acetylesterase HE

CoVs and toroviruses of the Coronaviridae family are specific for the O-Ac SA receptors. Their S and HE glycoproteins are similar to influenza C virus HEF. CoVs and all toroviruses bear HE gene form class I envelope membrane proteins of about 400 amino acid residues which bear 7 to 12 N-glycosylation sites [50]. HE multimer forms enter virions. Bovine CoV (BCoV) and HCoV-OC43, similar to influenza C virus, recognize Neu5,9Ac2 and bear SA-9-*O*-acetylesterase [8]. CoV HEs are all O-acetylesterases. The HE enzymes found in torovirus, CoV and influenza C virus are evolutionarily interspecies-mutated with about 30% homology by heterologous RNA recombination [51] and horizontal gene transfer. Therefore, viral HEs are diverse and widespread over evolution.

### 5.2. Substrate Diversity of the CoV HEs

HEs as envelope proteins are found in CoVs, orthomyxoviruses and toroviruses. Coronaviral HEs are involved in virus attachment to SA species. HE protein in β-CoVs binds to Neu5,9Ac2 form SA and agglutinates the red blood cells (RBCs) of rodents [52]. As with SA-*O*-acetylesterase, HE potentiates viral entry with the S protein and spreading via the mucosal glycans. It contains a carbohydrate-recognizing domain (CRD) known in lectin. The HE glycan-binding domain (GBD) mediates virus attachment to SAs on host cells. HE is the only HA. This indicates that compared to the S glycoprotein, HE is only minor a HA and the S glycoprotein mainly attaches to the cell surface. The HE protein of murine hepatitis virus (MHV), an enveloped CoV, binds to SA-4-acetylester or SA-9-*O*-acetylester of the carcinoembryonic antigen cell adhesion molecule 1a (CEACAM; known as CD66a) as the key receptor [53]. Murine CoVs HEs acquired by horizontal gene transfer, bind to C9-*O*-Ac Neu5Ac. However, some murine CoV HEs cannot bind to C4-*O*-Ac Neu5Ac. The original mouse MHV HE binds to C9-*O*-Ac Neu5Ac, while the MHV S-strain HE evolutionarily acquired the ability to bind to C4-*O*-Ac Neu5Ac [12,53,54]. In terms of structure, the C5 N- and C9 O-Ac Neu5Ac-accomodating hydrophobic pocket was shifted to a C5 N- and C4-*O*-Ac Neu5Ac-accomodating pocket [55].

Type I HE is specific for the 9-*O*-acetylated SAs (9-*O*-Ac-SAs). Type II HE is specific for 4-*O*-Ac-SAs. The SA-binding shift indicates quasi-synchronous adaptations of the SA-recognition sites of the lectin and esterase domains. Type I HE monomers of β-CoV lineage A have a bimodular enzyme–lectin domain similar to cellular glycan/carbohydrate-modifying proteins. Originally, HE homologs are found in various viruses including toroviruses and orthomyxoviruses such as the influenza virus C/D and isavirus, as well as the exceptional case of β-CoV lineage A among CoVs. The HE gene was transmitted to a β-CoV lineage A progenitor via horizontal gene transfer from a 9-*O*-Ac-Sia–recognizing HEF, as shown in influenza virus C/D. HE acquisition and expansion occurred by cross-species transmission over HE evolution and this phenomenon reflects viral evolutionary adaptation to host SA-containing glycans. Therefore, CoV HE receptor switching precedes virus evolution driven by SA-containing glycan diversity of hosts. For instance, the BcoV HE prefers 7,9-di-*O*-Ac-SAs, which is also a target of the bovine torovirus HE. For a more outstanding case, such a switching event occurred in the murine CoVs for the β-CoV lineage A type switch toward O-Ac-SA recognition. In the HE specificity of murine CoVs, two different murine CoV subtypes of virus group exist with one subtype possessing the typical 9-*O*-Ac-SA (type I) attachment factor and the other exclusively 4-*O*-Ac-SA (type II) attachment virus group [56].

The first coronaviral HE proteins identified were from the porcine hemagglutinating encephalomyelitis virus (PHEV), BCoV and HCoV-OC43, which bear SA-9-*O*-acetylesterases similar to HEF [8]. Rat CoV (RCoV) has SA-4-*O*-acetylesterases, converting Neu4,5Ac2 to Neu5Ac [53,57,58]. Some murine CoVs prefer 4-*O*-Ac-SAs and others 9-*O*-Ac-SAs. HCoV-OC43 and BCoV prefer α2-6-SA 9-*O*-acetylation by their SA-*O*-acetyleseterases. The S glycoproteins of BCoV and HCoV-OC43 are Neu5,9Ac2-recognizing lectins and agglutinate murine, rat and chicken erythrocytes due to the enriched 9-*O*-Ac-SA species [52]. BCoV and HCoV-OC43 adapted to SA receptor determinants of 9-*O*-Ac-SA receptors [59]. For a second receptor, the binding of S glycoprotein to Neu5,9Ac2 receptor is essential for entry into cells. BCoV-infection is prevented by prior treatment of cells with NA enzyme or with viral SA-*O*-acetylesterases, blocking the roles of HE and S glycoprotein in SA-dependent entry to host cells.

## 6. CoVs Infection of Human Hosts

### 6.1. CoVs Utilize SAs and SA Linkages as Attachment and Entry Sites to Human Host Cells

Several β-CoV genera such as BCoV bind to O-acetylated SAs and bear an acetylesterase enzyme to act as a host cell RDE. Certain α-CoV and γ-CoV are deficient for the comparable acetylesterase enzyme but have a preference to NeuAc or NeuGc type SA species. Infectious bronchitis virus (IBV) and transmissible gastroenteritis virus are such examples. Additionally, both α-CoV and γ-CoV also include sub-members deficient of any SA-recognizing activity. During evolution, some subtypes of SARS-CoV and HCoV-229E acquired SA-binding capacity. The SA-binding activities of BCoV, transmissible gastroenteritis coronavirus (TGEV) and IBV are well known [60].

#### 6.1.1. α-Coronavirus

In α-CoVs such as TGEV, HA-activity is attributed to the SA-recognizing activity to α2,3-NeuGc [61,62]. The SA-binding site is present on the N-terminal region of the S-glycoprotein of TGEV. TGEV has two types with enteric and respiratory tropism. The respiratory TGEV has the porcine aminopeptidase N (pAPN)-binding domain and SA-binding domain. Nucleotide 655 of the S gene is essential for enteric tropism and the S219A mutation of the S glycoprotein confers the enteric to respiratory tropism shift. In addition, a 6-nucleotide insertional mutation at nucleotide 1124, which yields the Y374-T375insND shift of the S glycoprotein, causes enhanced enteric tract tropism. TGEV interacts with SA species on mucin-like glycoprotein (MGP), a highly glycosylated protein, in an SA-dependent manner, on mucin-secreting goblet cells [6]. MGP SA-binding allows virus entry via the mucus layer to the intestinal enterocytes. Different from TGEV, the S glycoprotein of porcine CoV has no hemagglutination activity due to deletion of the SA-binding site of the S glycoprotein [61]. The loss of SA-binding activity is correlated to the non-enteropathogenicity. SAs function as HA-mediated entry determinants for TGEV, causing the enteropathogenic outcome of the virus, and SA-recognition activity is also responsible for virus amplification in cells. SA-binding activity-deficient TGEV can propagate in cells through pAPN, known as CD13, as a receptor [62,63]. The SA-binding activity potentiates infection and is crucial for intestinal infection.

#### 6.1.2. β-Coronavirus

In β-CoV, HE mediates viral attachment to O-Ac-SAs and its function relies on the combined CBD and RDE domains. Most β-CoVs target 9-*O*-Ac-SAs (type I), but certain strains switched to alternatively targeting 4-*O*-Ac-SAs (type II). For example, the SA-acetylesterase enzyme in BCoVs and HCoV-OC43 is known to have hemagglutinizing activities as a type of SA-9-*O*-acetylesterase [8]. The SA-acetylesterase is the HE surface glycoprotein in BCoV. The three-dimensional structure of BCoV HE is similar to other viral esterases [9]. The HE gene is found only in the β-CoV genus. The acetylesterase of murine CoVs differs in its substrate binding specificity from that of BCoV and HCoV-OC43, which is specific for O-acetyl residue release from SA C-9. Murine CoVs prefer to esterize 4-*O*-acetyl-NeuAc [64]. The β-CoV acetylesterase destroys the receptors and this specificity is similar to that of influenza viruses. Acetylesterase activity can be inhibited by diisopropyl fluorophosphate and this agent decreases viral infection levels [65]. As deduced from the SA acetylesterase of HCoV-OC43 [8], the 9-*O*-Ac-SA species is a receptor binding determinant for erythrocytes and entry into cells [59]. The BCoV HE protein has dual activity of acetylesterase and HA [9]. BCoV widely agglutinates erythrocytes and purified HE only agglutinates Neu5,9Ac2-enriched erythrocytes of rats and mice. BCoV and HCoV-OC43 can agglutinate chicken erythrocytes, while purified HE cannot. In contrast to the HE protein, purified S glycoprotein can agglutinate chicken erythrocytes [52], indicating that the major HA is the S protein which acts as the major SA-binding protein. However, the role of O-Ac-SAs is not certain to be essential in receptors, and SA-binding activity may be essential only to the HE protein, but not to the S glycoprotein [54].

#### 6.1.3. γ-Coronavirus

In γ-CoVs, IBV strains, known as poultry respiratory infectious pathogens, can agglutinate erythrocytes. IBV prefers to recognize α2,3-NeuAc and the SA functions as a host entry receptor for infection [66]. Glycosylation of IBV M41 S1 protein RBD is crucial for interaction with chicken trachea tissue and RBD N-glycosylation confers receptor specificity and enables virus replication. The heavy glycosylated M41 RBD has 10 glycosylation sites. N-glycosylation of IBV determines receptor specificity. However, the host receptor has not yet been found. NA treatment reduces the binding of soluble S to kidney and tracheal epithelial cells. The IBV S protein recognizes epithelial cells in a SA-dependent manner. The SA-binding ability of IBV is necessary for infection of tracheal epithelial cells and lung respiratory epithelial cells [67]. The SA-binding site is located on S1 of the IBV S protein, although the IBV-specific protein receptor is not known. In contrast to BCoV or HCoV-OC43, IBV lacks an RDE. SA binding of IBV is likely more essential than in other viruses such as TGEV.

#### 6.1.4. Torovirus

In torovirus, which belongs to the family Coronaviridae, the toroviruses are grouped into the Torovirinae subfamily and the Torovirus genus. The known toroviruses can infect four species of hosts, constituting bovine, equine, porcine and human toroviruses. They mildly infect swine and cattle through the HE protein, which is similar to the β-CoV HE protein [68]. The HE protein is a class I membrane glycoprotein which forms homodimers with a MW of 65 kDa. The RDE protein HE reversibly binds to glycans [15] through binding to SAs. The acetyl-esterase activity disrupts SA binding. HE hemagglutinates mouse erythrocytes and cleaves the acetyl-ester linkage of glycans and acetylated synthetic substrate p-nitrophenyl acetate (pNPA) [69]. Similar to CoV, torovirus HE is an acetylesterase type, which cleaves the O-acetyl group from the SA C-9 position using Neu5,9Ac2 and N-acetyl-7(8),9-*O*-NeuAc [64]. However, torovirus HE exhibits a restricted specificity for the Neu5,9Ac2 substrate, but not for the Neu5,7(8),9Ac3 substrate, with a unique SA-binding site generated by a single amino acid difference in porcine Thr73 and bovine Ser64 for each HE [70].

### 6.2. SARS-CoV-2 Recognizes 9-*O*-Acetyl-SAs and MERS-CoV Recognizes α2,3-SAs as Attachment Receptors

The S glycoprotein SARS-CoV-2 initiates infection of the host cells. The molecular basis of CoV attachment to sugar/glycan receptors is an important issue, as demonstrated by recent cryo-EM defining the structure of the CoV-OC43 S glycoprotein trimer complexed with a 9-*O*-acetylated SA [56]. Cryo-EM structures of the trimeric ectodomain of S glycoprotein were observed using forms complexed with Neu5Ac, Neu5Gc, sialyl–Lewis^X^ (SLe^X^), α2,3-sialyl-N-acetyl-lactosamine (α2,3-SLacNAc) and α2,6-SLacNAc, respectively. The receptor-binding site is commonly conserved in all CoV S glycoproteins, which attach to 9-*O*-Ac-SA species with similar ligand-binding pockets to the CoV HEs and influenza virus C/D HEF glycoproteins, indicating conserved recognizing structures [25]. The S glycoprotein-9-*O*-acetyl-SA interaction resembles the ligand-binding pockets of CoV HEs and influenza virus C/D HE fusion glycoproteins. HCoV-OC43 and BCoV recognize 9-*O*-Ac-SA. S glycoproteins engage 9-*O*-acetyl-SAs. The 9-*O*-acetyl SAs are the binding site for HCoV-OC43 S glycoprotein and related β-1 CoV S glycoproteins, however SA-binding sites on the 9-*O*-acetyl sialyl receptors of MERS-CoV S glycoprotein and HCoV-OC43 S glycoprotein are different [71]. Thus, CoVs use two different entry and attachment receptors. Therefore, S glycoproteins of CoVs are distinct from influenza virus A HAs, which bind to the Neu5Ac species by conserved binding sites. The ligand-binding sites of BCoV HE enzyme, influenza HEF enzyme and CoV S glycoprotein have evolved 9-*O*-Ac-SA binding through hydrogen bonding with the 9-*O*-acetyl carbonyl group and hydrophobic pocket formation with the 9-*O*-acetyl methyl group [71,72]. However, influenza HA cannot bind to 9-*O*-acetyl-SAs but can bind to NeuGcs [73]. The HCoV-OC43 S glycoprotein, HCoV-HKU1 S glycoprotein, BCoV S glycoprotein and PHEV S glycoprotein, therefore, share the ligand-binding specificity of influenza C/D HEF enzyme, although they are functionally more similar to influenza virus A/B HA, whereas CoV HE or influenza virus A/B NA have RDE activities

CoV HEs are functionally similar to influenza virus C/D HEF glycoproteins. In CoV, the S glycoprotein recognizes the 9-*O*-Ac-SA sugar, while the HE acts as the RDE enzyme with SA-*O*-acetyl-esterase activity to release virions from infected host cells. For example, HCoV-OC43 also has a similar HE as an RDE [71]. In influenza C and D viruses, HEF glycoproteins act similarly to the CoV HE [74]. In influenza A virus, RDE NA releases virions from host cells. However, MERS-CoV does not have a similar enzyme and thus MER-CoV binding to SA receptors is mediated by energetically reversible interactions of the lipid rafts with increased SA receptors [75], thus enhancing dipeptidyl peptidase 4 (DPP4) or carcinoembryonic antigen-related cell adhesion molecule 5 (CEACAM5) recognition power and viral entry [76] and membrane-associated 78-kDa glucose-regulated protein (GRP78) [77].

MERS-CoV S glycoprotein can hemagglutinate human erythrocytes and mediates virus entry into human respiratory epithelial cells. MERS-CoV S glycoprotein attachment is not observed for 9-*O*-acetylated or 5-N-glycolyl SAs, but is observed for α2,3-SA linkage over α2,6-SA linkages. SA-binding sites of MERS-CoV S glycoprotein and HCoV-OC43 S glycoprotein are not conserved [78], although they engage α2,3-SAs on the avian host cell surface [79]. MERS-CoV recognizes α2,3-SA and to a lesser extent the α2,6-SAs and sulfated SLe^X^ for binding preference. Thus, S glycoproteins may have independently evolved SA recognition. The acquisition of SA-binding ability of MERS-CoV S seems to be an evolutionarily recent event, because HKU4 S1 and HKU5 S1 cannot hemagglutinate human erythrocytes [75], indicating flexible evolutionary exchange allowing cross-species transmission towards host cell tropism of CoVs. In conclusion, CoV recognition of 9-*O*-Ac-SAs for infection is based on a conserved sequence for engagement of SA-related carbohydrate ligands across CoVs and orthomyxoviruses.

### 6.3. Host Receptors of CoVs

CoV S spikes recognize diverse surface molecules as the attachment or entry site. Animal and human coronaviruses evolve to acquire the same host receptors and attachment factors and overcome the interspecies barrier from animals to human. Specifically, S glycoprotein interaction with its binding receptor determines host tropism, pathogenicity and therapeutic clues [80]. CoVs recognize multiple host receptors via distinct S domains. The host receptors for β-CoV SARS-CoV includes angiotensin-converting enzyme 2 (ACE2). As a lineage C β-CoV, the MERS-CoV S glycoprotein binds to DPP4 [81,82,83]. MERS-CoV S glycoprotein recognizes α2,3-SA over α2,6-SA-bearing receptors. The N-terminal subunits of the S1/S1A/S1B/S1D complex of MERS-CoV recognize DPP4. MERS-CoV recognizes CEACAM5 as the attachment factor for entry [78]. Among the six HCoVs, the α-CoV HCoV-229E S protein recognizes human APN (hAPN) [84]. α-CoV HCoV-NL63 and the lineage B β-CoV SARS-CoV S glycoproteins bind to ACE2. Meanwhile the protein receptors specific for lineage A β-CoVs such as HCoV-HKU1 and HCoV-OC43 are not known yet.

BCoV, HCoV-OC43, HCoV-HKU1 and TGEV recognize O-acetyl-SAs as attachment molecules. In addition to O-acetyl-SA, HCoV-HKU1 spikes additionally bind to major histocompatibility complex class I (MHC-I) C as attachment sites [85]. SARS-CoV uses dendritic cell (DC)-specific intercellular adhesion molecule (ICAM)-3–grabbing nonintegrin (DC-SIGN) for attachment [86]. For glycan interaction, HCoV-NL63 and mouse hepatitis virus utilize heparan sulfate (HS) proteoglycans as attachment enhancers [87]. In general, ACE2, APN, heat shock protein A5 (HSPA5), furin, heparan sulfate proteoglycans (HSPGs) and O-acetyl-SA are CoVs-recognizing candidates.

#### 6.3.1. Angiotensin-Converting Enzyme 2 (ACE2) as the SARS-CoV Host Receptor

##### Structure and Role of the Host SARS-CoV Receptor ACE2

SARS-CoV-2 needs ACE2 for entry. Host proteases such as human ACE2 help viral entry through removement of a barrier to enter human cells through unknown receptors. Human ACE2 is known for its role as the SARS-CoV-2 entry receptor and the SARS-CoV receptor. The enzyme ACE-2 in the renin-angiotensin system (RAS) is associated with CoV entry into lungs. ACE2 mediates SARS-2002 entry into host cells via S glycoprotein interaction with the ACE2 receptor. The ACE2 levels on the plasma membrane correlate with virus infectivity. ACE2 expression is present in most tissues such as the lung epithelium. It is highly expressed by respiratory epithelial cells and type I/II lung alveolar epithelial cells [88]. The host receptor is not linked to the classification of CoVs. MERS-CoV, a β-CoV, does not recognize the ACE2 receptor. In contrast, the α-CoV HCoV-NL63 recognizes the ACE2 receptor. ACE2 is a membrane-anchored carboxypeptidase with 805 amino acid residues and is captopril-insensitive. It contains 17 amino acid residues as a signal peptide in the N-terminal region, a type I membrane-anchored domain in the C-terminal region, an extracellular N-terminal domain with heavy N-glycans, a N-terminal SARS-CoV-binding and carboxypeptidase site and a short C-terminal cytoplasmic tail. The ACE2 gene is located on chromosome Xp22. Two ACE2 forms are known, a membrane-bound form and a soluble form.

ACE cleaves angiotensin I (Ang I) substrate to Ang II. Ang II recognizes the Ang II receptor type 1 (AT1R), contributing to systemic and local vasoconstriction, fibrosis and salt retention in vascular organs. ACE2 has the opposite function of ACE. ACE2 is a close homolog to human ACE. ACE2 activity on Ang II is about 400-fold higher than that on Ang I. Ang-1 to Ang-7 recognize the G protein-coupled receptor (GPCR) Mas to activate vasorelaxation, cardioprotection, antioxidative action, antiinflammation and anti-Ang II-signaling. Therefore, the ACE2-Ang-1 to Ang-7 axis is a target candidate for cardiovascular diseases. ACE2 shows similar binding structures between nCoV and SARS-CoV. The three proteins of ACE, Ang II and AT1R contribute to progression of lung injury in humans. ACE2 removes a single amino acid residue from Ang II to yield the vasodilator, named Ang 1-Ang 7. ACE2 cleaves Ang-I to Ang 1–Ang 9 and Ang II to Ang-1 to Ang-7. The biggest difference between ACE2 and ACE is that ACE2 has a non-inhibitory property by ACE inhibitors.

Pulmonary ACE2 is potentially a candidate target in CoV-involved inflammatory pathogenesis. If ACE inhibitors and Ang II-AT1 blockers are dosed, ACE2 expression is increased. However, currently we have no conclusive evidence that the inhibitors help SARS-CoV or SARS-CoV-2 entry. Rather, SARS-CoV infection reduces ACE2 expression. Therefore, SARS-CoV-2 host tropism is not related to ACE2 expression. ACE2 levels and ANG II/ANG 1–7 levels regulate the pathogenic progression. ACE2 expression is upregulated by gene polymorphisms and ACE inhibitors or Angiotensin II receptor blockers such as sartans.

##### Host Cell ADAM17 and TMPRSS2 Competitively Cleave ACE2

A disintegrin and metallopeptidase domain (ADAM) family of Zn-metalloproteinases belongs to membrane proteins. The well-known ADAM17 is a TNF-α-converting enzyme (TACE), called the sheddase for TNF-α. Other ADAM sheddase family members include ADAM9, ADAM10 and ADAM12. ADAM17 mediates ACE2 shedding. SARS-CoV S glycoprotein activates cellular TACE and consequently facilitates virus entry. Soluble ACE2 as the N-terminal carboxypeptidase domain form is derived from the original ACE2 form by an ADAM17 metalloprotease in the membrane [89]. ADAM17 is indeed an enzyme that can convert membrane type pro-TNF-α to soluble TNF-α, a functional proinflammatory cytokine. Therefore, ADAM17 inhibition indicates an anti-inflammatory response and ADAM17 inhibitors are promising candidates for TNF-α-induced inflammatory diseases. The short C-terminal domain of ACE2 is removed by ADAM17 and TMPRSS2. However, TMPRSS2 cleaves ACE2 competitively with the ADAM17 metalloprotease. SARS-S protein-ACE2 binding leads to ADAM17/TNF-α-converting enzyme (TACE)-cleavage of ACE2, facilitating extracellular ACE2 shedding and consequent SARS-CoV entry into host cells [90,91]. Only TMPRSS2 cleavage allows SARS-CoV entry into host cells through endocytosis and fusion. Soluble ACE2 also recognizes the virus and prevents SARS-CoV-2 infection. SARS-CoV-2 infection requires membrane ACE2 and TMPRSS2. The ACE2–B0AT1 complex binds to the S glycoprotein of SARS-CoV-2. Intestinal membrane ACE2 and lung TMPRSS2-shedded ACE2 can act as alternative entry sites for SARS-CoV-2. SARS-CoV-2 infects the lungs and intestine via TMPRSS2-cleaved ACE2. If TMPRSS2 is engaged in SARS-CoV-2 entry and ACE2 downregulation, TMPRSS2 inhibition would lead to COVID-19 prevention. Although ACE2 is expressed both in type I and type II lung alveolar epithelial cells, SARS-CoV and SARS-CoV-2 target only type II epithelial cells due to the ACE2–TMPRSS2 interaction. Therefore, supplementation of ACE2 (soluble ACE2) or Ang-1 to Ang-7 should be a way to reduce SARS-CoV-2-related symptoms.

TMPRSS2-cleaved ACE2 is involved in SARS-CoV and MERS-CoV infections. SARS-CoV-2 uses ACE2 for cell entry through TMPRSS2 priming of the S glycoprotein (Figure 7). Infection of the H7N9 influenza and H1N1 influenza A subtype viruses are also mediated by TMPRSS2-cleaved ACE2. This implies that TMPRSS2 can be targeted as a strategic antiviral therapy [92]. Transmembrane protease serine 2, termed TMPRSS2, a type II TM Ser protease (TTSP), also cleaves ACE2. The human TMPRSS2 gene, located on chromosome 21, comprises androgen receptor elements (AREs) in the upstream 5′-flanking region [93]. TMPRSS2 expression is regulated in an androgen-dependent manner. The TMPRSS2 gene encodes 492 amino acids. The original form is cleaved into the major membrane form and the minor soluble form. TMPRSS2 activates protease activated receptor 2 (PAR-2) and activated PAR-2 upregulates matrix metalloproteinase-2 (MMP-2) and MMP-9. TMPRSS2-activated hepatocyte growth factor (HGF) induces c-Met receptor signaling. TMPRSS2 activates SARS-CoV and MERS-CoV. The SARS-CoV S glycoprotein is cleaved by host-borne TMPRSS2, human airway trypsin-like protease (HAT), TM protease, serine 13 (MSPL), serine protease DESC1 (DESC1), furin, factor Xa and endosomal cathepsin L/B. SARS-CoV can enter cells upon cleavage by protease TMPRSS2 or endosomal cathepsin L/B [90]. Virus S protein precursor is cleaved by host proteases. The spikes are cleaved by endosomal cathepsin and by Golgi or plasma membrane TMPRSS2 in the step of assembly or attachment and release. The serine protease inhibitor camostat effectively blocks lethal SARS-CoV infection to mice. However, serine protease and cathepsin inhibitors are not effective. Thus, TMPRSS2 is suggested to be an acting protease for SARS-CoV entry into host cells, but not by cathepsin. Cis-cleavage liberates SARS-CoV S glycoprotein fragments into the extracellular supernatant. Trans-cleavage activates the SARS-CoV S glycoprotein on the target cells, potentiating efficient SARS-CoV S glycoprotein-driven viral fusion. TMPRSS2-activated SARS-CoV facilitates enveloped virus entry into cells. TMPRSS2 is important for SARS-CoV entry and infection [81,94,95,96].

The fact that SARS- and MERS-CoV infections are potentiated by TMPRSS2 indicates that TMPRSS2 is a promising target for therapeutic agents. For example, several Ser protease inhibitors such as camostat mesylate inhibit TMPRSS2–ACE2-involved SARS-CoV-2 entry. camostat, a serine protease inhibitor, reduces influenza virus titers in cell culture. camostat-treated TMPRSS2 inhibition in Calu-3 cells greatly reduces SARS-CoV viral titers and improves survival rate in SARS-CoV infected mice. A treatment of 10-μM camostat blocks MERS-CoV entry to African green monkey kidney (Vero)-TMPRSS2 cells and blocks viral RNA synthesis in Calu-3 cells upon MERS-CoV infection. Aprotinin is a polypeptide with 58 amino acid residues that was isolated from bovine lungs. Another serine protease inhibitor, nafamostat, inhibits MERS-CoV entry and infection by TMPRSS2 inhibition [93]. nafamostat mesylate blocks the TMPRSS2–ACE2-involved SARS-CoV-2 envelope–PM fusion and prevents SARS-CoV-2 entry [95]. nafamostat mesylate inhibits viral entry and thrombosis in COVID-19 patients. Similarly, an FDA-approved mucolytic cough suppressant, Bromhexine hydrochloride (BHH), inhibits TMPRSS2 (IC50 0.75 μM) and hence blocks infection of CoV and influenza virus. MPRSS2 as a host factor plays a pivotal role in SARS-CoV and MERS-CoV infections. FDA-approved TMPRSS2 inhibitors are yet under development. Because TMPRSS2 mediates efficient viral entry and replication, it should be a promising target for new therapeutics against CoV infection.

#### 6.3.2. Dipeptidyl peptidase-4 (DPP4) as MERS-CoV Receptor

The Ser exopeptidase DPP-4/human CD26 (PDB: 4L72), a type II TM ectopeptidase, functions as a host cell receptor for MERS-CoV. The RBD structure was characterized by crystallography approaches of the MERS-CoV S glycoprotein–DPP4 complex. DPP4 is a single type II TM glycoprotein with a small cytoplasmic tail in the N-terminal region and is present as a homodimeric form. DPP4 cleaves X-proline dipeptides from the N-terminal region. S glycoprotein recognizes SA species and DPP44 as the attachment and entry receptors, respectively. The MERS-CoV S1 N-terminal domain attaches to DPP4 as the host receptor [81]. The S2 C-terminal domain of MERS-CoV anchors to cellular PM to enter. MERS-CoV S glycoprotein is cleaved at a sequence between the S1 and S2 domains [96]. Another cleavage site S2′ is present in the S2 domain. MERS CoV S glycoprotein sialyl receptors are expressed in the camel nasal respiratory epithelial cells and the human lung alveolar epithelial cells, which express DPP4. Binding capacities are hindered by the SA 9-*O*-acetyl group or SA 5-N-glycolyl group [75].

#### 6.3.3. CEACAM Receptor

Entry of host cells needs binding of S glycoproteins to the CEACAM receptor, forming S-protein-mediated membrane fusion. The trimeric S glycoprotein bears three S1 receptor heads. The three S1 heads of the virus bind to three receptor molecules on the host cell. Cholesterol is indirectly involved in membrane fusion through CEACAM engagement into “lipid raft” microdomains, increasing multiple S protein interaction with the receptors and triggering membrane fusion [97]. The enveloped CoV, MHV, binds to CEACAMs on cholesterol-depleted cells in BHK cell cultures. The NTD of S1 recognizes CEACAM1. For MERS-CoV, another CEACAM5 isoform is the attachment factor for virus entry [75]. The CoV S1 NTD has a similar tertiary structure to human galactose-recognizing galectins. MHV S1 NTD binds murine CEACAM1a and BCoV S1 NTD binds sugar [98,99,100]. CEACAM1a is a cell adhesion protein (CAM) and its mRNA is alternatively spliced. The cryo-EM structure of MHV S complexed with CEACAM1a was elucidated [101]. Thus, HCoVs evolutionarily combined the galectin gene of hosts into their S1 glycoprotein gene, while BCoV S1 protein is present without such gene recombination but contains the sugar-recognizing lectin capacity. MHV S1 protein also evolutionarily acquired murine CEACAM1a-recognizing activity [102]. Therefore, CoVs are under evolution to adapt their host receptor interaction to infect cross-species hosts [80,103]. On the host side, to escape the lethal pressure from CoV infections, hosts have also evolved to acquire SA-binding proteins such as siglecs to inhibit or activate the innate immune cells.

Both raft and non-raft CEACAMs are involved in the virus–cell membrane fusion event. Formation of CEACAM-associated MHV particles or CEACAM-induced MHV fusion is possible by GPI-anchored CEACAMs through the binding between CEACAM and S proteins. However, MHV can bind to both GPI- and TM-anchored CEACAMs. In addition, soluble CEACAMs also mediate S glycoprotein-driven fusion [104]. This implies that membrane anchors are not intrinsically necessary. In fact, CEACAMs are present in different tissue-specific isoforms [105]. Nevertheless, GPI-anchored CEACAMs are more effective for MHV infection than TM-anchored CEACAMs. Soluble CEACAM receptors can bind to viral S glycoproteins and induce conformational shifts to acceptable S glycoprotein-involved membrane fusions [106]. For example, soluble CEACAM forms interacts with S1 fragments [107] and alters the S1–S2 association stability [108] and S1 oxidation confirmation [109]. S proteins are structurally shifted prior to membrane fusion. For the cross-linking of viruses and cells, integral hydrophobic peptides of the S2 chain are embedded into membranes via membrane hydrophobic cholesterols.

#### 6.3.4. Membrane-Associated 78-kDa Glucose-Regulated Protein (GRP78) or HSPA5

MERS-CoV S glycoprotein also recognizes a 78-kDa glucose–regulated protein (GRP78) or heat shock 70 kDa protein 5 (HSPA5), known as binding immunoglobulin protein (BiP) or Byun1, which is encoded by the HSPA5 gene in humans. HSP5A is a ER-resident unfolded protein response (UPR) protein. Stressed cell status such as viral infection increase expression and translocation of HSPA5 to the PM to form a membrane protein complex. GRP78 modulates MERS-CoV entry in the presence of the DPP4 as a host cell receptor. Additionally, lineage D β-CoV and bat CoV HKU9 (bCoV-HKU9) also bind to GRP78 [76]. A cell surface receptor, GRP78, was predicted to be another COVID-19 receptor as an S glycoprotein binding site [110]. The prediction was made using the combined technology of molecular modeling docking with structural bioinformatics. GRP78 or BiP is a chaperone protein located in the ER lumen [111]. Known ER-bound enzymes include activating transcription factor 6 (ATF6), inositol-requiring enzyme 1 (IRE1) and protein kinase RNA (PKR)-like ER kinase (PERK) [112]. Depending on threshold of unfolded protein accumulation, GRP78 releases IRE1, ATF6 and PERK, and is activated, resulting in translation inhibition and refolding. Stress-overexpressed GRP78 can avoid ER retention and is translocated to the membrane. GRP78 translocated to the cell PM can recognize viruses by its substrate-binding domain (SBD) for virus entry into the cell (Figure 8). In sequence and structural alignments and protein–protein docking, RBD of the CoV spike protein recognizes the GRP78 SBDβ as the host cell receptor. The predicted region III (C391–C525) and region IV (C480–C488) of the S glycoprotein and GRP78 are highly potential binding sites. Region IV is the GRP78 binding-driving force. These nine amino acid residues are being molecularly targeted for the designation and simulation of COVID-19-specific drugs. This process is the mechanism underlying the cell surface HSPA5 (GRP78) exposure and this is exploited to be used for pathogen entry. Such pathogenic entry into host cells has been observed in multiple infections including pathogenic human viruses such as human papillomavirus, Ebola virus, Zika virus and HcoVs—as well as fungal *Rhizopus oryzae* [113,114,115,116]. Therefore, natural products can inhibit cell-surface HSPA5 recognition of the viral S glycoprotein.

#### 6.3.5. Aminopeptidase N (APN) is a Receptor of α-CoV HCoV-229E

Among the six HCoVs, the α-CoV HCoV-229E S protein recognizes hAPN known as CD13 or membrane alanyl aminopeptidase (EC 3.4.11.2). Porcine epidemic diarrhea coronavirus virus (PEDV) binds to protein receptor APN of human- and pig NeuAc species as its co-receptor. Apart from hAPN, TGEV and PEDV bind to SA species [117], although SA recognition by TGEV is not essential in the first step of entry cycle. HCoV-229E recognizes hAPN known as CD13 for its entry receptor. hAPN (PDB: 4FYQ) or CD13 (EC 3.4.11.2), which is a Zn-dependent metalloprotease, has a MW 150 kDa with 967 amino acids. CD13 is a type II TM protein with a short cytoplasmic domain in the N-terminal region and long extracellular region in the CTD. The CTD has a pentapeptide sequence specific for the Zinc–MMPs. The APN binding domain is located on the CTD of PEDV S1 (amino acid 477–629 residues), while the SA-binding domain is found in the N-terminal region of PEDV S1 (amino acid 1–320 residues) [118]. CD13 is also a receptor for HCoV-229E, human cytomegalovirus, porcine CoV TGEV, feline infectious peritonitis virus (FIPV), feline enteric virus (FeCV) and canine-infectious CoVs [119,120,121,122]. Homodimeric CD13 digests luminal peptides. The hAPN-encoding ANPEP gene is a dominant component in proximal tubular epithelial cells, small intestinal cells, macrophages, granulocytes and synaptic membranes. If this gene is defective, leukemia or lymphoma are transformed [123]. Porcine and human APN exhibit about 80% protein identity. FIPV and FeCV are in the same group as HCoV-229E and TGEV. Thus, porcine APN is also an attachment site for pig TGEV with an additional second receptor. HCoV-229E first binds to CD13 and consequently clusters CD13 in caveolae-associated lipid rafts [120].

#### 6.3.6. Heparan Sulfate (HS) is the HCoV-NL63 Attachment Site

For glycan interaction, HCoV-NL63 and MHV utilize heparan sulfate proteoglycans (HSPGs) as attachment enhancers [87,124]. Viruses recognize HSPGs as attachment molecules. In the spike (S) protein-deficient virions, the M protein recognizes HSPG. The S proteins generally bind to the viral cellular receptor. However, the M protein also acts as a receptor in the early step of HCoV-NL63 infection. The M membrane protein of HCoV-NL63 recognizes the attachment site of HSPGs. HCoV-NL63 M protein binds to HSPG for the initial attachment of virus to host cells and thereafter, the M and S proteins cooperate for virus entrance into the host cells [125]. HSPGs are glycosaminoglycan (GAG)-carrying proteins frequently used as a secondary receptor for viral entry. HSPGs are composed of covalent-bonded HS chains as a GAG form. The HS GAG linkage structure of tetrasaccharide exhibits GluAβ1,3GlcNAcα1,4Galβ1,3Galβ1,4Xylβ-*O*-serine. Glycosyltransferases involved in HS GAG synthesis include GlcAT-II (glucuronosyltransferase) and GlcNAcT-II (N-acetylglucosaminyltransferase II) for heparan sulfate synthesis (Figure 9). GAG is used as docking sites for virus interaction with the host cell surface. GAGs contain negatively charged N- and O-sulfated sugars [126]. The biosynthetic pathway and biologic roles in early embryogenic morphogenesis and vulval morphogenesis of HS and chondroitin sulfate GAG have been elucidated in *Caenorhabditis elegans* [127]. The negative charges mediate the interaction of GAGs and their ligands through electrostatic forces. Interaction of HSPG with ligands potentiates many virus infectious cycles. For examples, adeno-associated virus, human T cell lymphotropic virus type 1, human papilloma virus 16, herpes viruses, hepatitis B and C viruses, Kaposi’s sarcoma-associated herpesvirus, human papilloma viruses and Merkel cell polyoma virus recognize the HSPGs [128,129]. HSPGs increase virulence upon interaction with viral factors required for viral attachment and replication.

#### 6.3.7. Major Histocompatibility Complex Class I (MHC-I) C is an Attachment Site for HCoV-HKU1

Although HCoV-HKU1 utilizes O-acetyl-SAs as attachment sites, the HCoV-HKU1 S protein also interacts with MHC-I C (HLA-C) as an additional attachment molecule [85].

#### 6.3.8. DC-SIGN (CD209) is a Binding Candidate for SARS-CoV Entry

SARS-CoV uses the C-type lectins of DC-SIGN and DC-L-SIGN as additional or secondary receptors. Glycans on the S glycoprotein are recognized by DC/L-SIGN for virus attachment and entry. Seven glycosylation sites of the S glycoprotein have been found to be essential for DC/L-SIGN-driven virus entry [86,130].

#### 6.3.9. Tetraspanin CD9 is a Surface factor for MERS-CoV Entry Via Scaffold Cell Receptors and Proteases

Tetraspanin CD9, but not tetraspanin CD81, associates with DPP4 and the type II TM serine protease (TTSP) member TMPRSS2, a CoV-activating protease, to form a cell surface complex [131]. This CD9–DPP4–TMPRSS2 complex permits MERS-CoV pseudovirus entrance into the host cells. The tetraspanins have four TM spanning regions linked by one large and one small loop in the extracellular region. Tetraspanins form virus entry baselines and open CoV entry routes. To help viral entry into host cells, MERS-CoV S interacts with DPP4 receptors via the RBD. Receptor involvement causes cleavage using proteases such as the previously described TMPRSS2. Association of tetraspanin CD9 with the DPP4–TMPRSS2 complex triggers the S glycoprotein. MERS-CoVs enter the cells via endocytosis and cathepsins cleave the S proteins [132].

### 6.4. Effects of Receptor and Ligand S Glycosylation on Virus–Host Interaction

SAs are predominant surface determinants for pathogen attachment, adherence and entry to host cells. Eleven representative vertebrate virus families utilize SAs as initial entry receptors or as attachment factors. Interaction of virus with SA-containing glycans is complex because virus SA-binding lectins are inherently of very low affinity. Viruses acquire enzymes to catalyze virion elution by regional depletion of binding receptors [56]. TM S glycoprotein recognizes oligosaccharide receptors. Using cryo-EM technology and observed structures of S glycoprotein trimers of CoV OC43 complexed with 9-*O*-acetylated SA, S glycoprotein was demonstrated to mediate virus adhesion and entry to host cells. All CoV S proteins show conservation in binding to 9-*O*-acetyl-SAs. MERS-CoV also recognizes 9-carbon sugar SA species. MERS-CoV S-1A binds to SA species. For example, SAα2,3- over SAα2,6-linkages expressed in human erythrocytes and mucins are preferentially targeted by MERS-CoV S-1A. Binding is hence blocked by SA modification to 5-N-NeuGc and 7, 9-*O*-NeuAc species [73]. For example, impairment of ACE2 receptor glycosylation does not influence S-glycoprotein-ACE2 interaction, however, SARS-CoV-2 virus entry into respiratory epithelial host cells was downregulated [133]. Changes in ACE2 N-glycans do not apparently influence interaction with the SARS-CoV S glycoprotein, but instead, impair viral S glycoprotein-mediated membrane fusion. The receptor glycan structures decide the entry of some human viruses. Changes in ACE2 receptor sialylation influences interaction affinity between virus ligands and host receptor. Inter-species or individual genetic variations such as drift and mutation may occur in SARS-CoVs. This explains currently emerging differences in CoV responses within the same population such as humans.

On the other hand, from the aspect of virus ligand, the S glycoprotein decorates viral surfaces and is, therefore, the target for vaccination design. Virus internalization requires potential glycosylation of viral glycoproteins. Among the three viral envelope components, S and M are the major glycoproteins and E is nascent and not glycosylated. The M glycoprotein consists of a short glycosylated ectodomain in the N-terminal region. The S glycoprotein expressed in hemagglutinating encephalomyelitis virus is an HA that recognizes N-acetyl-9-*O*-NeuAc as a binding receptor expressed on erythrocyte surfaces [134]. For example, BCoVs attach to the surface receptor of N-acetyl-9-*O*-NeuAc (9-*O*-acetylated SAs) on host cells. TGEV and PEDV are currently known as a similar class of such CoVs. PEDV infects multiple hosts including bat, pig, human and monkey, where bats are considered to be the evolutionary origin for PEDV. The S glycoprotein of SARS-CoV-2 utilizes different glycosylation patterns to recognize its receptors. The glycosylation sites in minimal RBD exhibits similar sites to other CoVs. The trimeric SARS-CoV-2 S glycoprotein is also highly glycosylated with 66 N-glycans, but a few O-glycans [135]. Glycosylation of S glycoproteins leads to immune evasion. In the MERS-CoV and the bat-specific CoV-HKU4, glycosylation is linked to zoonotic infection for fusion-based entry [136].

## 7. Pharmacology of Glycan-Related Anti-SARS-CoV-2 Agents

The emerging CoV-pandemic requires therapeutic agents to block the recognition, binding, replication, amplification and propagation of the CoV in humans. Protease inhibitors, RNA synthase inhibitors and S2 inhibitors are potential targets, and several agents are currently being evaluated. Efficient therapeutic drugs are the most reliable option for patients. The first attachment step of the viral amplification cycle is initiated on the respiratory cell surfaces, driven by the viral S protein. This is a potential therapeutic target. Soon after the SARS-CoV-2 outbreak initiated, the CoV S glycoprotein was demonstrated to recognize ACE2 as a binding receptor on human cells. Human TMPRSS2 enzyme influences the CoV S glycoprotein activation, to facilitate virus infection. ACE2 binding and TMPRSS2 activation facilitate the CoVs to attach to human host cells. Mouse, nonhuman primate and human cells have been analyzed using single-cell RNA new generation sequencing (NGS). For example, for human infection, CoVs can enter nasal goblet secretory cells, because these cells express the proteins required for SARS-CoV-2 infection. In the lungs, the proteins are stored in the alveoli like air sacs of type II pneumocytes. In the intestine, the two proteins are expressed in entero-epithelial cells. ACE2 gene expression correlates with the IFN-related genes [137]. ACE2 helps lung cells to tolerate cellular damage. Therefore, CoVs may evolutionally take advantage of the defense mechanisms of host cells, hijacking such host-borne proteins.

In SARS-CoV-2, the ACE2 receptor is an attachment, entry and infection receptor into the cell, when the S glycoprotein is cleaved by a specific serine protease. SARS-CoV-2 infection is regulated by glycosylated SARS-CoV-2 viral particles and glycosylated ACE2 in the lung epithelial cells. RBD of the CoV S glycoprotein recognizes ACE2 [82]. Amino acid residues 442, 472, 479, 480 and 487 located on the receptor-binding motif (RBM) of the S glycoprotein RBD recognize human ACE2 [138]. Trimeric viral S glycoprotein is glycosylated and cleaved by a protease, furin, into two subunits, S1 and S2. Subunit S1 is further cleaved into the SA and SB domains and the SB domain recognizes human ACE2. The N-glycosylated S2 subunit is involved in virus-ACE2 complex formation [139]. Therefore, the glycosylated ACE2 receptor is a key molecule for virus binding and fusion.

Plasma sera prepared from infected patients is an alternative medication. The WHO has suggested this trial since the 2014 Ebola epidemic and 2015 MERS-CoV outbreak. In addition, Mab therapy is another option. For example, LCA50 Mab mimics produced by modification of plasma antibodies isolated from MERS-CoV patients was valuable [140]. Low molecular molecules are being examined for anti-virus activities from alkaloids, glycan derivatives and terpenoids. Recently, anti-CoV drugs are being approached using molecular modeling, docking and simulation methods. Computation-assisted drugs via molecular modeling and docking toward drug targets are applied as anti-viral compounds against CoVs. They target human ACE2, PLpro (PDB: 3e9s), the CoV main proteinase (PDB: 6Y84), 3-chymotrypsin-like (3C-like protease; 3CLpro), RdRp, helicase, N7 methyltransferase, human DDP4, RBD, protease cathepsin L, type II TM Ser protease or TMPRSS2. CoV 3CLpro (PDB: 6WX4) and the PLpro cleave the polyproteins to assemble virus proteins. For newborn RNA genomes, RdRp is used as a replicase for the complementary RNA strand synthesis, which uses the virus RNA template.

### 7.1. N-Glycosylation Inhibition by Chloroquine (CLQ) and Hydroxychloroquine (CLQ-OH)

CLQ and CLQ-OH are under investigation worldwide to treat COVID-19 (Figure 10). CLQ and its derivative CLQ-OH block CoV replication, amplification and spread in in vitro culture via inhibition of ACE2 receptor glycosylation. In HCoVs, interaction of the S glycoprotein with gangliosides initially occur as the first entry step during the replication cycle of the virus. CLQ and CLQ-OH have been alternative drugs for RA and several autoimmune diseases for 70 years, although they are anti-malaria prophylaxis drugs. CLQ-OH is an aminoquinoline with less toxicity than CLQ. CLQ-OH bears an N-hydroxyethyl side chain, which increases its solubility compared to CLQ [141]. CLQ-OH modulates activated immune cells via downregulation of TLR signaling and IL-6 production [142]. Clinical trials are also under consideration for the efficacy and safety of these drugs. Regarding the action mechanism(s), CLQ and CLQ-OH-mediated inhibition of ACE2 terminal glycosylation was considered. In in vitro Vero E6 cells, CLQ significantly inhibits SARS-CoV spread by interfering with ACE2 function, acting at the entry and post-entry steps of SARS-CoV-2 replication and infection. The binding affinity of ACE2 to S glycoprotein is simulated to be lowered by treatment with CLQ-OH or CLQ. CLQ may modify the binding affinity between ACE2 and S glycoprotein by alterations in ACE2 glycosylation or modification. CLQ-OH (EC50 0.72 μM) and CLQ (EC50, 5.47 μM) inhibit SARS-CoV-2 [143].

Using computer simulation techniques, CLQ and CLQ-OH have been suggested to recognize the enzymatic active site of the UDP-GlcNAc 2-epimerase, known as an essential enzyme in SA biosynthesis [144], blocking the sialylation of host cells. The mechanism underlying the glycosylation inhibition may support the antiviral properties of CLQ and CLQ-OH through interactions of CLQ or CLQ-OH with NDP-saccharide mutases or glycosyltransferases [145]. CLQ was reported to inhibit quinone reductase 2 [146], known as a catalytic mimetic or structural neighbor of UDP-GlcNAc 2-epimerases [147,148]. If CLQ or CLQ-OH inhibits SA synthesis, the inhibitory properties may support the antiviral activity of CLQ or CLQ-OH against SARS-CoVs because the SARS-CoV receptor ACE2 contains SA species. In fact, CLQ exhibits in vitro anti-SARS-CoV-1 activity via defective glycosylation of viral ACE2 in Vero cells [149]. In addition. the interference of CLQ or CLQ-OH with SA synthesis may broadly be applicable as an antiviral because the HcoVs or other orthomyxoviruses also utilize SAs as entry molecules [150]. However, the detailed mechanisms should be further elucidated. The CLQ treatment efficacy in Covid-19 patients has, however, not been conclusively determined.

### 7.2. Interaction of Membrane Gangliosides in Lipid Rafts with CLQ and CLQ-OH

Lipid rafts are also viral attachment sites. Viruses such as IBV, dengue virus, Ebola virus, hepatitis C virus, HIV, human herpes virus 6, measles virus, Newcastle disease virus, poliovirus, West Nile virus, foot-and-mouth disease virus, simian virus 40, rotavirus, influenza virus and Marburg virus also use lipid rafts for virus entry [151,152,153,154,155,156]. In avian CoV IBV, structural proteins of the IBV virus are co-localized with PM lipid rafts embedded with the ganglioside GM1. HCoV-229E entry is prevented by cholesterol depleted conditions because HCoV-229E clusters in caveolae-associated lipid rafts [157]. Caveolae of caveolin-1, -2 and -3 are cross-linked [158] and control the molecular distribution between rafts and caveolae in a regulatory mechanism. S protein-CD13 cross-linking occurs via CD13-caveolin-1 sequestering. HCoV-229E particles similarly exhibit a longitudinal distribution property. HCoV-229E-colocalized caveolin-1 undergoes the next step of virus infection. Caveolin-1 knockdown inhibited HCoV-229E endocytosis and entry and thus caveolin-1 is essential for HCoV-229E infection. TGEV also endocytoses by a clathrin-mediated mechanism in MDCK cells [159]. Other viruses including HCoV-OC43 also use an entry receptor sequestered to cross-linked caveolae [160]. In SARS-CoV, the first entry step to host cells needs ACE2 in intact lipid rafts by the S glycoprotein [151]. ACE2 is associated with caveolin-1 and GM1 in membrane rafts depending on its cell-type specific localization [161]. Raft integrity with cholesterol and ACE2 is necessary for SARS-CoV pseudovirus entry into Vero E6 cells and for SARS-CoV-microdomain-based entry. C-type lectin, CD209 L (L-SIGN), can also form lipid rafts and acts as a SARS-CoV receptor [162]. Information of the CoV entry pathways is important for therapeutic designation of SARS-CoV-targeting drugs, for example, if agents disrupt lipid-raft localization of the ACE2 receptor.

CLQ binds the SAs and gangliosides in lipid rafts with a high affinity. Therefore, CLQ or CLQ-OH prevents the S glycoprotein–ganglioside binding. CLQ (or CLQ-OH) binding to SA consequently prevents S glycoprotein binding to host receptors. The N-terminal region of SARS-CoV-2 S glycoprotein interacts with gangliosides. A ganglioside-binding site (GBS) or ganglioside-binding domain (GBD) is present in the NTD of the S glycoprotein of SARS-CoV-2. Using molecular modeling and simulation technology, CLQ has been suggested to recognize the SAs and gangliosides. Human type Neu5Ac binds to CLQ and CLQ-OH. Thus, SAs are binding targets of CLQ and CLQ-OH. CLQ and CLQ-OH have two specific recognition sites in the polar sugar residues of ganglioside GM1. The first site is found at the tip of the sugar residues of GM1 with an interaction energy of −47 kJ/mol. The CLQ rings face the GalNAc residue of GM1, while the second site is in a large region of the sugar-ceramide junction and the sugar residues. Several amino acid residues of the S protein NTD, which are Phe-135, Asn-137 and Arg-158, recognize the ganglioside GM1. The S glycoprotein NTD-GM1 complex is suggested to form a trimolecular complex with two molecules of ganglioside GM1 anchored to the NTD of S protein [163]. The ACE2-binding RBD is suggested to be a potential GBS located on a differential site of the S glycoprotein NTD. The protein sequence interfacing surface of the NTD is the consensus GBDs [164]. The amino acids Gly, Pro and/or Ser residues found in GBD motifs are in the same 111–158 amino acids of the NTD as the ganglioside-attachment interface. The GBD is conserved throughout viral isolates from worldwide COVID-19 patients. The GBD potentially increases viral attachment ability to PM lipid rafts and contact between host ACE-2 and S protein [165]. The interaction between CLQ-OH and 9-*O*-acetyl-NeuAc is also similar to the 9-*O*-acetyl-NeuAc-CLQ interaction. The CLQ-OH OH group enhances the interaction of CLQ with SA via a hydrogen bond [163]. In conditions with CLQ or CLQ-OH derivative treatment, the S glycoprotein cannot bind to gangliosides in in silico studies, which are used to uncover the action mechanism. CLQ and CLQ-OH prevent the binding of S glycoprotein to gangliosides. The CLQ-SA complex is formed in a mixed surface and balls by the positioning of the negative charged COOH group of Neu5Ac and one of the two cationic charges of CLQ [163]. CoVs preferentially bind to 9-*O*-acetyl-NeuAc [60], differentiating with other viral properties.

As CLQ interacts with the GM1 sugar part, the N-terminal domain of the S protein loses viral attachment capacity to the cell receptors [166]. The S protein NTD and the CLQ/CLQ-OH maintain the same position during GM1 binding, consequently preventing GM1 binding to the S protein and the drug at the same time, because the NTD and the CLQ/CLQ-OH simultaneously recognize GM1. Asn-167 forms a hydrogen bond with the GalNAc residue, whereas an aromatic Phe-135 stacks to the Glc residue of GM1. Therefore, the antiviral activities of CLQ and CLQ-OH is to block the interaction between the SARS-CoV-2 S glycoprotein and gangliosides on host cell surfaces. The lipid composition of host cell PM can also be a potential target for preventive and therapeutic drugs against such viruses.

## 8. Conclusions

The SARS-CoV-2-caused COVID-19 pandemic is a global public health issue in the 21st century. In order to coordinate efforts against and mitigate the public health consequences of the spreading and outbreak of the disease, the international community has been exchanging independent information and knowledge. The scientists and epidemiologists exchange COVID-19 information to highlight interdisciplinary approaches. The pandemic outbreak issue has raised interest in the pathology and epidemiology of the disease. The current COVID-19 pandemic resulted in establishment of the COVID Action Platform of the World Economic Forum (WEF) to perform evidence-based cutting-edge research and analyze the fast-evolving pandemic. The Virus Outbreak Data Network (VODAN) of the GoFair Data Alliance also pursues to apply the best remedy against the pandemic infection. In the aspect of basic biology, most β-CoVs recognize 9-*O*-acetyl SAs, but certain viruses have switched to binding 4-*O*-acetyl SA. Originally, HE is found in other viruses such as toroviruses and orthomyxoviruses (influenza C/D and isavirus). The exceptional β-CoV lineage A is the only one to bear HE among the CoVs. Virus entry and replication inhibitors need to be urgently developed. Computation-fused artificial intelligence accelerates therapeutic agent development. The SARS-CoV-2 genome shows 80% similarity to the previous SARS-CoV and SARS-targeting agents can be commonly treated to the related SARS-CoV-2 patients. For better understanding of the entry pathway of SARS-CoV-2, the importance of the carbohydrates including SAs on cell and virus surfaces is again emphasized.

## Figures and Tables

**Figure 1 ijms-21-04549-f001:**
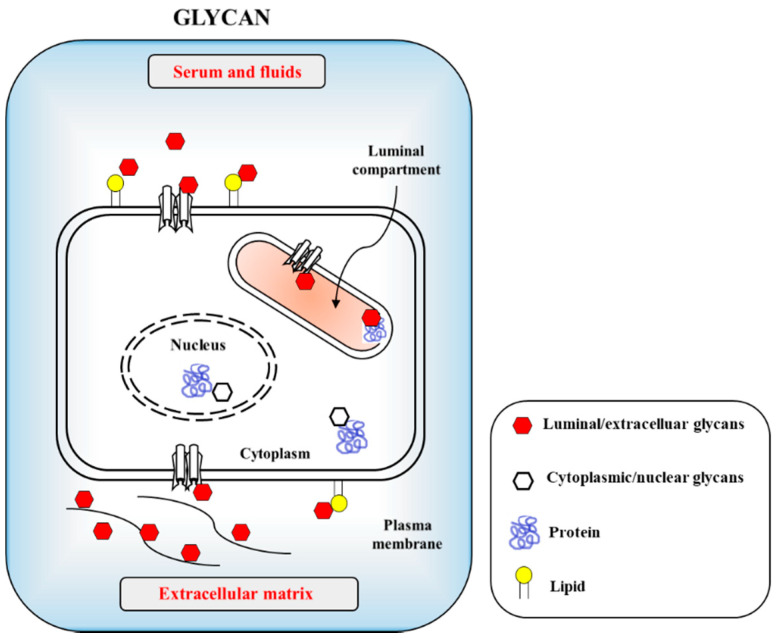
Cellular glycans are recognized by infectious agents including viruses and bacteria through protein–carbohydrate interaction (PCI) or lectin–carbohydrate interaction (LCI). The carbohydrates are uses as cellular adhesion sites in eukaryotic cells. Host cell surfaced and cytosolic glycans include glycoproteins, glycolipids and proteoglycans with minor glycan species of O-GlcNAc present in nucleus and cytosols.

**Figure 2 ijms-21-04549-f002:**
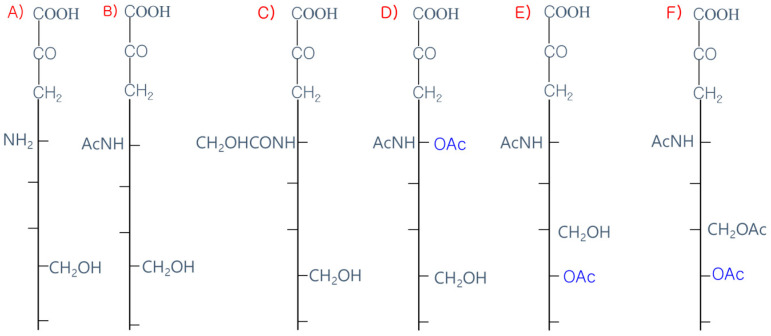
Diverse structures of sialic acids (SA). (**A**) Neuraminic acid; (NeuC); (**B**) N-acetyl neuraminic acid (NeuAc); (**C**) N-glycolyl neuraminic acid (NeuGc); (**D**) N, O-diacetyl neuraminic acid (occurs in horse); (**E**) N, O-diacetyl neuraminic acid (occurs in bovine); (**F**) N-acetyl O-diacetyl neuraminic acid (occurs in bovine).

**Figure 3 ijms-21-04549-f003:**
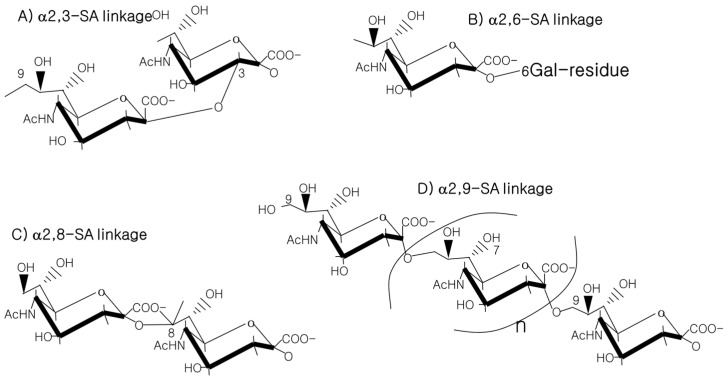
SA linkages of α2-3, α2-6, α2-8 or α2-9 to the SA or Gal residues.

**Figure 4 ijms-21-04549-f004:**
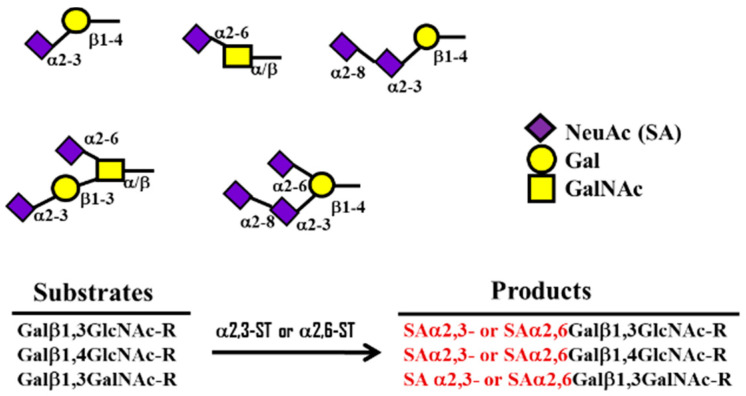
Formation of α2,3 ST or α2,6 SA structures by α 2,3- and 2,6-sialyltransferase (ST) using substrates such as Galβ-1,4-GlcNAc.

**Figure 5 ijms-21-04549-f005:**
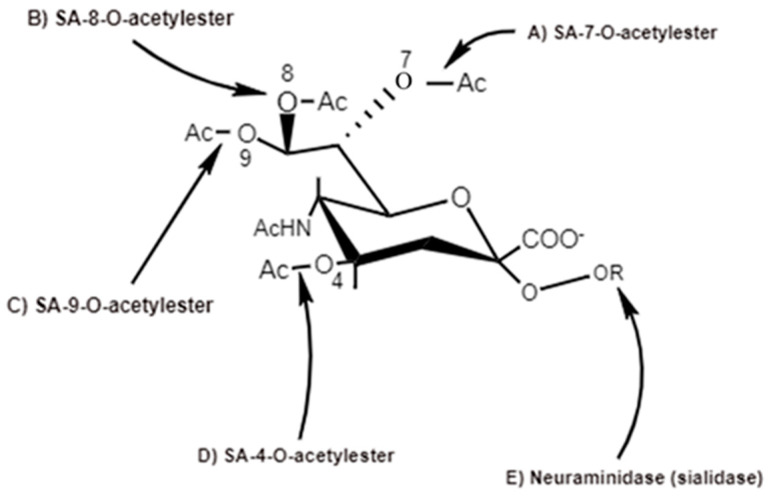
Action sites of viral SA-*O*-acetylesterases (C4, C7, C8 and C9) specific for 4-*O*-SA-, 7-*O*-SA-, 8-*O*-SA and 9-*O*-SA and neuraminidases [6].

**Figure 6 ijms-21-04549-f006:**
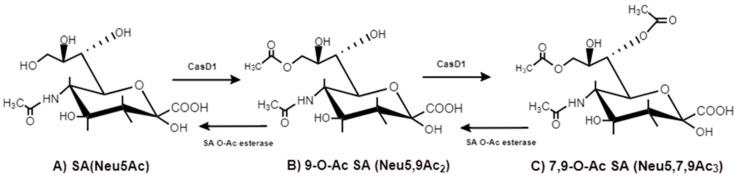
CasD1, SA O-acetyltransferase, transfers acetyl groups to C7 position of SA (Neu5Ac), from which it migrates to the C9 position (Neu5,9Ac2). The additional acetyl group is added to C7 of SA (Neu5,7,9Ac3) by the same CasD1. The SA O-acetylesterase cleaves of the acetyl groups.

**Figure 7 ijms-21-04549-f007:**
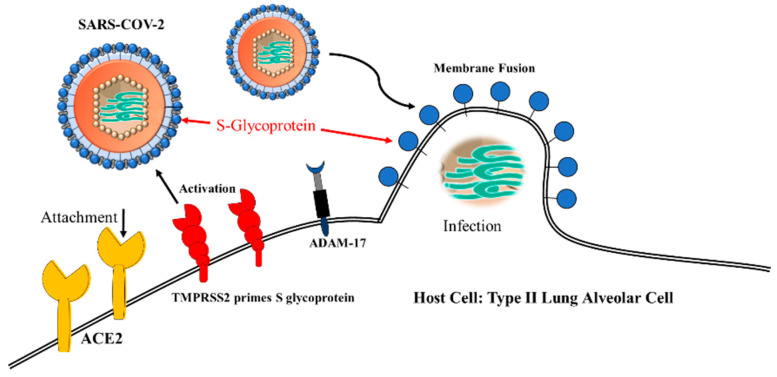
SARS-CoV-2 uses angiotensin-converting enzyme 2 (ACE2) for cell entry through TMPRSS2 priming of S glycoprotein.

**Figure 8 ijms-21-04549-f008:**
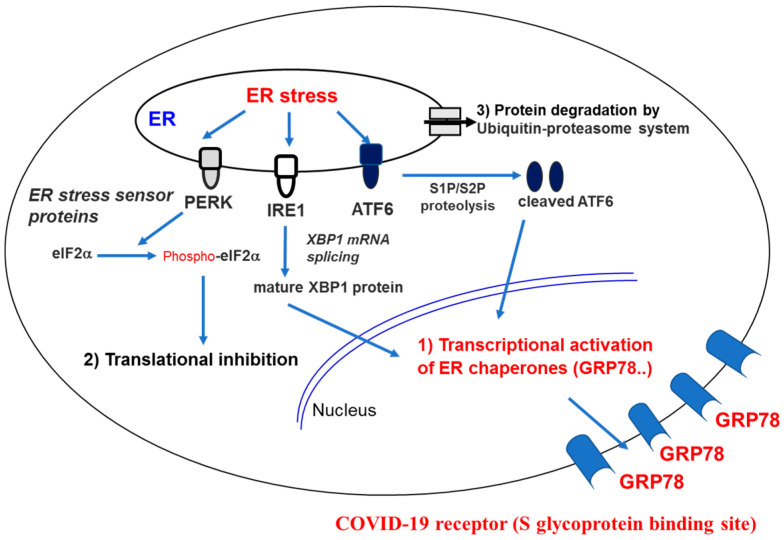
Endoplasmic-reticulum (ER) stress responses and COVID-19 receptor (S glycoprotein binding site). Balance of ER stress and unfolded protein response (UPR) are regulated. ER-stress sensor proteins are IRE1 (inositol requiring 1), ATF6 (activating TF 6) and PERK (PKR-like ER kinase).

**Figure 9 ijms-21-04549-f009:**
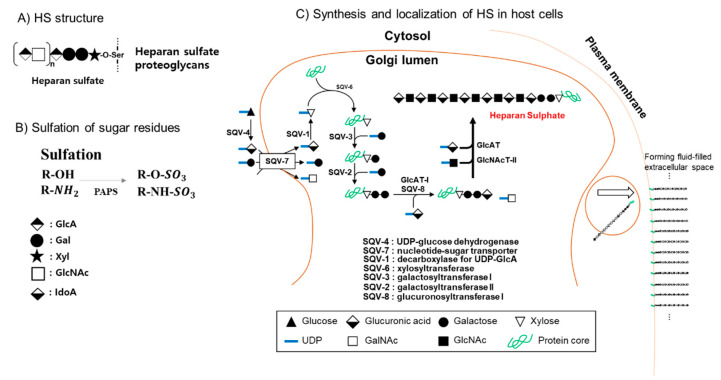
Structure of glycosaminoglycan (GAG)-linkage tetrasaccharide, GluAβ1,3Galβ1,3Galβ1,4Xylβ-*O*-serine and HS; (**A**) HS structure; (**B**) Sulfation of sugar residues; (**C**) Synthesis and localization of HS in host cells. Glycosyltransferases involved in GAG synthesis include (i) GlcAT-II (glucuronosyltransferase) and (ii) GlcNAcT-II (N-acetylglucosaminyltransferase II) for heparan sulfate synthesis. β-D-glucuronic acid (GlcA), α-L-iduronic acid (IdoA) and 2-*O*-sulfo-α-L-iduronic acid (IdoA(2S) are composed [124].

**Figure 10 ijms-21-04549-f010:**
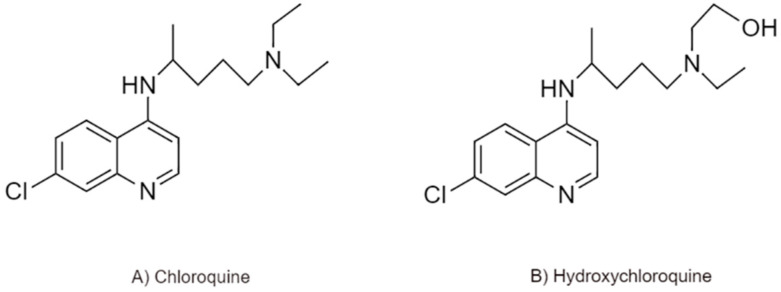
Structures of (**A**) CLQ and (**B**) CLQ-related CLQ-OH as predicted UDP-GlcNAc 2-epimerase inhibitors.

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
