# Peer review of "SARS-CoV-2 Evolutionary Adaptation toward Host Entry and Recognition of Receptor O-Acetyl Sialylation in Virus–Host Interaction"

_ijms, 2020, doi:10.3390/ijms21124549_

Round 1

Reviewer 1 Report

Reviewer report:

The manuscript summarizes most of the potential Covid-19 protein targets that could be promising for therapeutic intervention.  The authors discuss rather unexploited mechanisms such as the interaction of O-acetylated sialic acids that interact with lectin-like Spike glycoprotein of SARS CoV-2 as the initial attachment of the viruses to enter to the host cells.  The authors also summarized the available data gathered for previous coronaviruses as well as for influenza viruses (e.g. hemagglutinin-esterease enzyme (HE)) and pointed out possible analogous mechanisms that could be exploited for Covid-19 drug discovery.  There are several recently published papers that discuss the major proteins – protein interaction between the virus and the hosts and the potential druggable protein targets.  One of the well-cited recent papers that are missing from the references referred 67 druggable human proteins or hosts that are targeted by 69 existing FDA-approved drugs.  (Gordon, David E., et al. "A SARS-CoV-2-human protein-protein interaction map reveals drug targets and potential drug-repurposing." BioRxiv (2020)). https://www.biorxiv.org/content/biorxiv/early/2020/03/23/2020.03.22.002386.full.pdf.

Interestingly, hemagglutinin-esterease enzyme (HE) and the interactions with sialic acids as key elements in the initial virus attachment are not among the listed druggable targets in the above referred paper (Figure 1b), thus, the present account provides an important addendum and offers novel opportunities for therapeutic intervention. 

Question:

There are certain statements in the manuscript regarding the potential role of Chloroquine (CLQ) or hydroxychloroquine (CLQ-OH) in the initial attachment through binding to sialic acid components, while other recent papers describe different mechanism of action for those agents.

The authors note: “From computer simulation technique, CLQ and CLQ-OH are suggested to recognize the enzymic active site of the UDP-GlcNAc 2-epimerase, known as an essential enzyme in the SA biosynthesis [178], blocking the sialylation of host cells.” “Chloroquine (CLQ) binds SAs and gangliosides with high affinity. Therefore, CLQ or hydroxychloroquine (CLQ-OH) prevents the S glycoprotein-ganglioside binding. CLQ (or CLQ-OH) binding to SA consequently prevent the S glycoprotein binding to host receptors. “ “In the modelling simulation, CLQ and CLQ-OH recognize SA residues.

The above paper (David E., et al. ) proposes a different mechanism for CLQ and CLQ-OH for their antiviral effect. “SARS-CoV-2 Nsp6 protein interacts with the Sigma receptor, which is thought to regulate ER stress response. Chloroquine, which is currently in clinical trials for COVID-19 and has 109 nM activity vs the Sigma1 receptor, and low μM activity against the Sigma2 receptor. Because many patients are already treated with drugs that have off-target impact on Sigma receptors, associating clinical outcomes accompanying treatment with these drugs may merit investigation, a point to which we return.” (see Figure 5a, and Table 1b in the referred paper…)

Could you please comment on these different assumptions?

Other remarks:

Figure 6 and 7. Use standard ChemDraw for structure drawing to improve quality.

https://www.perkinelmer.com/category/chemdraw

Literature: what are the numbers in the square brackets?  It is probably an incorrect insertion or formatting of the references…

Author Response

Reviewer report: (Reviewer 1)

The manuscript summarizes most of the potential Covid-19 protein targets that could be promising for therapeutic intervention.  The authors discuss rather unexploited mechanisms such as the interaction of O-acetylated sialic acids that interact with lectin-like Spike glycoprotein of SARS CoV-2 as the initial attachment of the viruses to enter to the host cells.  The authors also summarized the available data gathered for previous coronaviruses as well as for influenza viruses (e.g. hemagglutinin-esterease enzyme (HE)) and pointed out possible analogous mechanisms that could be exploited for Covid-19 drug discovery.  There are several recently published papers that discuss the major proteins – protein interaction between the virus and the hosts and the potential druggable protein targets.  One of the well-cited recent papers that are missing from the references referred 67 druggable human proteins or hosts that are targeted by 69 existing FDA-approved drugs.  (Gordon, David E., et al. "A SARS-CoV-2-human protein-protein interaction map reveals drug targets and potential drug-repurposing." BioRxiv (2020)). https://www.biorxiv.org/content/biorxiv/early/2020/03/23/2020.03.22.002386.full.pdf.

Interestingly, hemagglutinin-esterease enzyme (HE) and the interactions with sialic acids as key elements in the initial virus attachment are not among the listed druggable targets in the above referred paper (Figure 1b), thus, the present account provides an important addendum and offers novel opportunities for therapeutic intervention. 

Answers

   As commented, the SARS-CoV-2 human host interacting proteins are recently suggested for the referred 332 SARS-CoV-2- human protein-protein interactions (PPIs). 67 druggable human proteins or host factors are raised as targets for development of antiviral therapeutics against SARS-CoV-2” was described in Abstract. Also, in Introduction, “Recently, 332 protein candidates have been suggested to be SARS-CoV-2-human protein interacting proteins through the PPIs. Then, 66 human proteins as druggable host factors have further been characterized for possible FDA-approvable drugs [10]” was newly inserted. The reference has been newly cited in the revision.

[10] Gordon DE, David E, et al. 2020. A SARS-CoV-2-Human Protein-Protein Interaction Map Reveals Drug Targets and Potential Drug-Repurposing. BioRxiv (2020). https://www.biorxiv.org/content/10.1101/2020.03.22.002386v1. (A SARS-CoV-2 protein interaction map reveals targets for drug repurposing. Nature. 2020 Apr 30. doi: 10.1038/s41586-020-2286-9).

Question:

There are certain statements in the manuscript regarding the potential role of Chloroquine (CLQ) or hydroxychloroquine (CLQ-OH) in the initial attachment through binding to sialic acid components, while other recent papers describe different mechanism of action for those agents.

The authors note: “From computer simulation technique, CLQ and CLQ-OH are suggested to recognize the enzymic active site of the UDP-GlcNAc 2-epimerase, known as an essential enzyme in the SA biosynthesis, blocking the sialylation of host cells.” “Chloroquine (CLQ) binds SAs and gangliosides with high affinity. Therefore, CLQ or hydroxychloroquine (CLQ-OH) prevents the S glycoprotein-ganglioside binding. CLQ (or CLQ-OH) binding to SA consequently prevent the S glycoprotein binding to host receptors. “In the modelling simulation, CLQ and CLQ-OH recognize SA residues.

The above paper (David E., et al. ) proposes a different mechanism for CLQ and CLQ-OH for their antiviral effect. “SARS-CoV-2 Nsp6 protein interacts with the Sigma receptor, which is thought to regulate ER stress response. Chloroquine, which is currently in clinical trials for COVID-19 and has 109 nM activity vs the Sigma1 receptor, and low μM activity against the Sigma2 receptor. Because many patients are already treated with drugs that have off-target impact on Sigma receptors, associating clinical outcomes accompanying treatment with these drugs may merit investigation, a point to which we return.” (see Figure 5a, and Table 1b in the referred paper…)

Question: Could you please comment on these different assumptions?

The answer:

I appreciate for the careful setting of the CLQ and CLQ-OH interaction with the SAs. As pointed out, the above paper (David E., et al. ), which I herewith cited it, the CLQ and CLQ-OH exhibit their antiviral effects. The interaction between the SARS-CoV-2 Nsp6 and host Sigma receptor induces ER stress response. However, CLQ modulates the Sigma1 receptor (higher concentration required) and Sigma2 receptor (lower μM activity). Clinical patients administered with CLQ or CLQ-OH exhibit no direct target potentials on Sigma receptors, claiming still uncertain action mechanistic outcomes. The issue of CLQ and CLQ-OH in efficacy is still controversial because there is no direct evidence by molecule to molecule (it is just like ligand-receptor concept). Although the reviewer mentioned the ER stressed response mediated by CLQ and CLQ-OH, there are many suggestions for their activities rather than orthodox mechanism. The present manuscript also deal with the the ER stressed response by several agents with the unknown mechanism.

During the revision, I have rewritten with additional backgrounds of the CLQ or CLQ-OH and SA synthesis enzymes. In 6.1.1. N-Glycosylation inhibition by chloroquine (CLQ) and hydroxychloroquine (CLQ-OH), the paragraph has been rewritten with the new citations.

“From computer simulation technique, CLQ and CLQ-OH are suggested to recognize the enzymic active site of the UDP-GlcNAc 2-epimerase, known as an essential enzyme in the SA biosynthesis [145], blocking the sialylation of host cells. The mechanism underlying the glycosylation inhibition may support the antiviral potentials of CLQ and CLQ-OH through interactions of CLQ or CLQ-OH with NDP-saccharide mutases or glycosyltransferases [146]. CLQ was reported to inhibit quinone reductase 2 [147], known as a catalytic mimetic or structural neighbour of UDP-GlcNAc 2-epimerases [148,149]. If CLQ or CLQ-OH inhibits the SA synthesis, the inhibitory potentials may support the antiviral activity of CLQ or CLQ-OH against SARS-CoVs because the SARS-CoV receptor ACE2 contains the SA species. In fact, the CLQ exhibits in vitro anti-SARS-CoV-1 activity via a defected glycosylation of viral ACE2 in Vero cells [150]. In addition, the interference of CLQ or CLQ-OH with SA synthesis can broadly be applicable for antiviral spectrum because the HcoVs or other orthomyxoviruses utilize SAs as entry molecules [151]. However, the detailed mechanisms should further be elucidated. The current conclusion of CLQ in Covid‐19 patients has however not been made through the world.

[146] Devaux CA, Rolain JM, Colson P, Raoult D. New insights on the antiviral effects of chloroquine against coronavirus: what to expect for COVID-19?. Int J Antimicrob Agents. 2020 May;55(5):105938. doi: 10.1016/j.ijantimicag.2020.105938.

[147] Kwiek JJ, Haystead TA, Rudolph J. Kinetic mechanism of quinone oxidoreductase 2 and its inhibition by the antimalarial quinolines. Biochemistry. 2004;43:4538–4547.

[148] National Center for Biotechnology Information MMDB—Entrez's Structure Database. http://www.ncbi.nlm.nih.gov/Structure/MMDB/mmdb.shtml (accessed Dec 14, 2005).

[149] Varki A. Sialic acids as ligands in recognition phenomena. FASEB J. 1997;11:248–255.

[150] Vincent M.J., Bergeron E., Benjannet S., Erickson B.R., Rollin P.E., Ksiazek T.G. Chloroquine is a potent inhibitor of SARS coronavirus infection and spread. Virol J. 2005;2:69.

[151] Olofsson S., Kumlin U., Dimock K., Arnberg N. Avian influenza and sialic acid receptors: more than meets the eye? Lancet Infect Dis. 2005;5:184–188.

Other remarks:

Question: Figure 6 and 7. Use standard ChemDraw for structure drawing to improve quality.https://www.perkinelmer.com/category/chemdraw

The answer: Thank you for your information of the ChemDraw site. As suggested, the ChemDraw program has been used to draw the better quality.

Fig. 6. has been modified according to the program.

Fig. 7. has also been modified according to the program.

Question: Literature: what are the numbers in the square brackets?  It is probably an incorrect insertion or formatting of the references…

  The answer: I am sorry for this uncareful preparation. As suggested, the styling has been made through the text and references.

Reviewer 2 Report

The Manuscript !SARS-CoV-2 host entry and receptor O-acetyl glycosylation and its ligand S glycosylation on virus-host interaction! is rumbling and difficult to read. 

I have some suggestions:

1- could you please simplify the manuscript and summarize it by highlighting the most important parts regarding the topic of Manuscript?

2- could you please reduce sections 6.1 and 6.2?

3- could you please improve paragraph 7 on therapies?

4- could you please review the bibliography and standardize it?

Author Response

Reviewer report: (Reviewer 2)

Comments and Suggestions for Authors

The Manuscript !SARS-CoV-2 host entry and receptor O-acetyl glycosylation and its ligand S glycosylation on virus-host interaction! is rumbling and difficult to read. 

I have some suggestions:

   The answer: As pointed out, I have revised to better form.

1- could you please simplify the manuscript and summarize it by highlighting the most important parts regarding the topic of Manuscript?

  The answer: As suggested, the length and conciseness have been made.

2- could you please reduce sections 6.1 and 6.2?

  The answer: As suggested, the length has been reduced.

3- could you please improve paragraph 7 on therapies?

  The answer: As suggested, the paragraph 7 has been improved and moved to the Conclusion section. This question is also raised by the Rev-4, as answered.

4- could you please review the bibliography and standardize it?

  The answer: I am sorry for this uncareful preparation. As suggested, the references have been rewritten.

Reviewer 3 Report

This manuscript addresses a range of topics relating to Coronavirus structure and binding and entry into cells, focussing primarily on the role of sialic acids. The organisation appears reasonable and I found the figures generally helpful, although figure 2 uses a scribble to represent “protein” while a later figure uses a similar scribble to represent RNA. I think the protein should be changed to an alternative shape as this is more consistent with common representation of proteins vs nucleic acids.

I think the scientific content of the review is probably mostly ok (barring some errors noted below), however the English requires substantial improvement to be certain, as the manuscript is hard work to read and the author’s meaning is sometimes unclear. There may well be additional errors. 

Line 105 - DNA should be RNA. Also this is an odd statement, as CoVs have a proof-reading mechanism, unlike most RNA viruses, so what exactly does the author mean?  

Both E and M are referred to as the most abundant proteins in the particle. 

Author Response

Reviewer report: (Reviewer 3)

This manuscript addresses a range of topics relating to Coronavirus structure and binding and entry into cells, focussing primarily on the role of sialic acids. The organisation appears reasonable and I found the figures generally helpful, although figure 2 uses a scribble to represent “protein” while a later figure uses a similar scribble to represent RNA. I think the protein should be changed to an alternative shape as this is more consistent with common representation of proteins vs nucleic acids.

I think the scientific content of the review is probably mostly ok (barring some errors noted below), however the English requires substantial improvement to be certain, as the manuscript is hard work to read and the author’s meaning is sometimes unclear. There may well be additional errors. 

The answer: Thank you for constructive comments. The Fig. 2 has been redrawn to meet the standard.

The modified Fig. 2.

Line 105 - DNA should be RNA. Also this is an odd statement, as CoVs have a proof-reading mechanism, unlike most RNA viruses, so what exactly does the author mean?  

The answer: I am sorry for this uncareful preparation. As suggested, the DNA has been corrected to RNA. The CoVs are positive (+) stranded single RNA viruses, capable to translate to proteins, functioning as mRNAs.

Both E and M are referred to as the most abundant proteins in the particle. 

The answer: I am sorry for this uncareful preparation. As suggested, the sentence has been corrected to “is an abundant structural protein with a MW 30 kDa”.

Reviewer 4 Report

Here are some minor comments for publication:

  1. In my opinion, Fig. 1 is completely unnecessary because the virus structure can be found in every textbook. In addition, in Fig. 1, the virus pattern is not compatible with the virus pattern in Fig. 8.
  2. When the abbreviation (e.g. HE or RDE, line 59) appears for the first time in a text, it should be explained.
  3. Paragraph between lines 82-89 is not logically related to the previous section. In my opinion, this fragment should be moved to other place in the manuscript.
  4. Protein names are sometimes in lowercase or sometimes in upper case.
  5. Scheme in Fig. 2 is unclear. The legend is missing for several abbreviations.
  6. In terms of graphics, the drawings in the review could be nicer and made using professional programs. In my opinion, they are unsightly and do not keep one style.
  7. Review submitted for review. Review paper normally is characterized by numerous references to literature. In the manuscript these references to literature are simply missing in many places. E.g. literature is required for the following sentences and in many other fragments of manuscript.

Line 48 The CoVs as the enveloped forms can also infect the gastrointestinal track (GIT), although most enteric viruses are naked in their morphology.

Line 49 CoVs can also infect rarely neural cells.

Line 51: Compared to influenza viruses, which selectively utilizes sialic acid (SA) linkages, this is sure.

Line 60: Most beta-CoVs target 9-O-acetylated SAs but certain species switched to recognizing 4-O-acetyl SA instead.

  1. Drawings of chemical structures cannot be adapted from Wikipedia but should be made by hand (see Fig. 10)
  2. In my opinion, all references to crystallographic data in the manuscript should contain PDB codes so that the reader can see the spatial structures.
  3. Papain-like protease abbreviations are different in the manuscript.
  4. Additional numbers appear unnecessarily in literature numbering at the beginning.

Author Response

Reviewer report: (Reviewer 4)

Comments and Suggestions for Authors

Here are some minor comments for publication:

  1. In my opinion, Fig. 1 is completely unnecessary because the virus structure can be found in every textbook. In addition, in Fig. 1, the virus pattern is not compatible with the virus pattern in Fig. 8.

The answer:

As suggested, the Fig. 1 has been deleted.

  1. When the abbreviation (e.g. HE or RDE, line 59) appears for the first time in a text, it should be explained.

The answer: I am sorry for this uncareful preparation. As suggested, The sentence of “the hemagglutinin-esterase (HE) enzyme, HE action relies on the typical carbohydrate-binding lectin action and receptor-destroying enzyme (RDE) domains” has been abbreviated for the Introduction section.

  1. Paragraph between lines 82-89 is not logically related to the previous section. In my opinion, this fragment should be moved to other place in the manuscript.

The answer: I am sorry for this uncareful preparation. As suggested, the paragraph has been moved to the Conclusion section.

  1. Protein names are sometimes in lowercase or sometimes in upper case.

The answer: I am sorry for this uncareful preparation. As suggested, the protein names have been corrected with consistency.

For example, “nsps” has been corrected to “Nsps”. Also many other words are the subject.

  1. Scheme in Fig. 2 is unclear. The legend is missing for several abbreviations.

The answer: Thank you for constructive comments. The same question has been raised by the Rev-3. The Fig. 2 has been redrawn to meet the standard. Also, the legend has been revised. Fig. 2. Cellular glycans are recognized by infectious agents including viruses and bacteria through PCI or LCI. The carbohydrates are uses as cellular adhesion sites in eukaryotic cells. Host cell surfaced and cytosolic glycans include glycoproteins, glycolipids and proteoglycans with minor glycan species of O-GlcNAc present in nucleus and cytosols.

  1. In terms of graphics, the drawings in the review could be nicer and made using professional programs. In my opinion, they are unsightly and do not keep one style.

The answer: I am sorry for this uncareful preparation. As suggested, all the figures have been redrawn using the https://www.perkinelmer.com/category/chemdraw

The same question has also been raised by the Rev-1, as answered above.

  1. Review submitted for review. Review paper normally is characterized by numerous references to literature. In the manuscript these references to literature are simply missing in many places. E.g. literature is required for the following sentences and in many other fragments of manuscript.

Line 48 The CoVs as the enveloped forms can also infect the gastrointestinal track (GIT), although most enteric viruses are naked in their morphology.

The answer: As suggested, the CoV infection of gastrointestinal track (GIT) is cited in [5].

[5] Lv X, Wang P, Bai R, Cong Y, Suo S, Ren X, Chen C. Inhibitory effect of silver nanomaterials on transmissible virus-induced host cell infections. Biomaterials. 2014 Apr;35(13):4195-203. doi: 10.1016/j.biomaterials.2014.01.054.

[6] Schwegmann-Wessels C, Zimmer G, Schröder B, Breves G, Herrler G. 2003. Binding of transmissible gastroenteritis coronavirus to brush border membrane sialoglycoproteins. J Virol. 77(21), 11846-8.

Line 49 CoVs can also infect rarely neural cells.

The answer: As suggested, the CoV infection of neural cells is cited in [7].

[7] Brison E, Jacomy H, Desforges M, Talbot PJ. Novel treatment with neuroprotective and antiviral properties against a neuroinvasive human respiratory virus. J Virol. 2014 Feb;88(3):1548-63. doi: 10.1128/JVI.02972-13.

Line 51: Compared to influenza viruses, which selectively utilizes sialic acid (SA) linkages, this is sure.

The answer: As suggested, the selective utilization of SA linkages is cited in [8].

   8] Schultze B, Wahn K, Klenk HD, Herrler G. 1988. Human and bovine coronaviruses recognize sialic acid-containing receptors similar to those of influenza C viruses. Proc Natl Acad Sci U S A. 85(12), 4526-9.

Line 60: Most beta-CoVs target 9-O-acetylated SAs but certain species switched to recognizing 4-O-acetyl SA instead.

The answer: As suggested, the most b-CoVs target 9-O-acetylated SAs but certain species switched to recognizing 4-O-acetyl SA instead [5,6] has been cited.

[5] Schultze B, Wahn K, Klenk HD, Herrler G. 1988. Human and bovine coronaviruses recognize sialic acid-containing receptors similar to those of influenza C viruses. Proc Natl Acad Sci U S A. 85(12), 4526-9.

[6] Schultze B, Wahn K, Klenk HD, Herrler G. 1991. Isolated HE-protein from hemagglutinating encephalomyelitis virus and bovine coronavirus has receptor-destroying and receptor-binding activity. Virology 180(1), 221-8.

  1. Drawings of chemical structures cannot be adapted from Wikipedia but should be made by hand (see Fig. 10)

The answer: As suggested, all the Figures have been redrawn using the above program https://www.perkinelmer.com/category/chemdraw

Fig. 10 is also the newly drawn figure, not copied.

  1. In my opinion, all references to crystallographic data in the manuscript should contain PDB codes so that the reader can see the spatial structures.SARS-CoV-2 spike glycoprotein (PDB: 6VSB)SARS-CoV-2 endoribonuclease (PDB: 6VWW).
  2. SARS-CoV-2 main protease (PDB: 6Y84)
  3. human angiotensin-converting enzyme 2 (ACE2) (PDB: 6M17).
  4. The answer: As suggested, the data have been strengthened with PDB codes.

SARS-CoV-2 RNA dependent RNA polymerase (PDB: 6M71)

SARS-CoV-2 3CL protease (3CL pro) (PDB: 6WX4)

SARS-CoV-2 NSP3 (PDB: 6VXS)

SARS-CoV-2 DPP-4/human CD26 (PDB: 4L72)

SARS-CoV-2 PLpro (PDB: 3e9s)

  1. Papain-like protease abbreviations are different in the manuscript. 
  2. The answer: I am sorry for this uncareful preparation. As suggested, the Papain-like protease abbreviations are corrected to the “PLpro”
  3. Additional numbers appear unnecessarily in literature numbering at the beginning. The title has been changed to cover the contents : SARS-CoV-2 evolutionary adaptation toward host entry and recognition of receptor O-acetyl sialylation in virus-host interaction
  4. The answer: I am sorry for this uncareful preparation. As suggested, the references are adjusted.

Round 2

Reviewer 2 Report

The Manuscript “SARS-CoV-2 evolutionary adaptation toward host entry and recognition of receptor O-acetyl sialylation in virus-host interaction” has been simplified and improved, as required.
The most important parts have been underlined.
I have no other comments.